# Unravelling Cocoa Drying Technology: A Comprehensive Review of the Influence on Flavor Formation and Quality

**DOI:** 10.3390/foods14050721

**Published:** 2025-02-20

**Authors:** Margareth Santander, Vanessa Chica, Hugo A. Martínez Correa, Jader Rodríguez, Edwin Villagran, Fabrice Vaillant, Sebastián Escobar

**Affiliations:** 1Corporación Colombiana de Investigación Agropecuaria (Agrosavia), Process & Quality Cocoa Laboratory, Centros de Investigación La Selva, Palmira, Central and Tibaitata—Km 14 Mosquera-Bogotá, Mosquera 250047, Colombia; msantander@agrosavia.co (M.S.); jrodriguezc@agrosavia.co (J.R.); fabrice.vaillant@cirad.fr (F.V.); 2Departamento de Ingeniería, Facultad de Ingeniería y Administración, Universidad Nacional de Colombia Sede Palmira, Palmira 763531, Colombia; vchicab@unal.edu.co (V.C.); hamartinezco@unal.edu.co (H.A.M.C.); 3Centre de Coopération Internationale en Recherche Agronomique pour le Développement—CIRAD, UMR QualiSud, 1, F-34398 Montpellier, France; 4UMR Qualisud, Univ Montpellier, CIRAD, Université d’Avignon, Université de la Réunion, Montpellier SupAgro, F-34000 Montpellier, France; 5Cacao of Excellence Programme, Bioversity International, 00118 Roma, Italy; s.escobar@cgiar.org

**Keywords:** Theobroma cacao, cocoa processing, cocoa drying methods, cocoa drying equipment, cocoa flavor quality, cocoa drying review

## Abstract

Cocoa quality serves as a differentiating factor that provides monetary and non-monetary benefits to farmers, defined by the genotype, agroecological conditions of cultivation, and the post-harvest processes involved in transforming seeds into cocoa beans, including harvesting, pre-conditioning, fermentation, and drying. Drying plays a crucial role in ensuring the sensory, chemical, and microbiological quality of the beans, as simultaneous mass and heat transfer phenomena occur during this process, along with chemical reactions (both enzymatic and non-enzymatic) that influence the concentration and dynamics of phenolic compounds, organic acids, methylxanthines, and the formation of volatiles, directly impacting flavor development in cocoa beans. This paper comprehensively reviews cocoa drying methods, variables, and equipment and analyzes their impact on these flavor-determining compounds. The findings highlight that drying significantly contributes to the production of differentiated and specialty quality traits. An integral relationship between the methods, operating variables, and drying equipment applied to cocoa and their implications for the volatile and non-volatile compounds is described.

## 1. Introduction

Cocoa has been widely used as an agricultural product for scientific research, as it is a food matrix that must undergo a set of physical and biochemical changes throughout post-harvest and processing operations to be processed from seeds into cocoa beans and chocolate. Chocolate consumption is mostly driven by the sensory pleasure caused by the perception of flavor (taste and aroma) [1]. The world cocoa market has considered flavor characteristics and established a categorization that includes the fine flavor cocoa (FFC) that exhibits specialty aromatic notes, such as fruity, nutty, caramel, and flowery, and, on the other hand, bulk cocoa, which has a strong basic cocoa flavor. The FFC has economic advantages because specialized markets offer premiums for high quality above the price offered on the stock exchanges. Therefore, the production of FFC is an alternative for cocoa farmers able to supply this type of market.

Cocoa transformation from seeds to beans includes four main operations: harvesting, preconditioning (in some cases, pre-drying and depulping), fermentation, and drying. Generally, most studies have highlighted fermentation as the main stage because, at this point, the endogenous seed components (carbohydrates and proteins) are degraded and metabolites (free amino acids, peptides, and reducing sugars) are formed [2]. These are known as flavor precursors, as in later stages such as drying and roasting, they interact and participate in chocolate flavor formation [3]. The quality of chocolate flavor is determined by multiple post-harvest processing stages, including fermentation, drying, and roasting, which collectively contribute to the development of key flavor precursors and characteristic aromatic compounds. Fermentation initiates a series of biochemical reactions that influence the sensory profile, while drying stabilizes these compounds and prevents the development of undesirable flavors [4,5]. Roasting further enhances the complexity of the flavor by promoting Maillard reactions and caramelization. If any of these processes are inadequate, the quality of the cocoa beans will be irreversibly compromised, leading to defects that cannot be corrected in subsequent stages, ultimately affecting the overall quality of chocolate and its derived products [6].

During the fermentation and drying of cocoa, complex chemical reactions occur, contributing to the development of flavor precursors essential for high-quality chocolate, as described by Santander et al. [2]. Fermentation initiates the breakdown of seed components catalyzed by endogenous enzymes. Proteolysis, mediated by aspartic endoproteases and carboxypeptidases, converts seed storage proteins, such as vicilin, into peptides and amino acids, key flavor precursors [7]. Simultaneously, invertases hydrolyze sucrose into reducing sugars, such as glucose and fructose, which later participate in Maillard reactions [8]. Polyphenol oxidase (PPO) catalyzes the oxidation of polyphenols, such as catechins, into quinones, reducing bitterness and astringency. Additionally, anthocyanins degrade into brown pigments, contributing to the characteristic color of fermented cocoa [9]. During drying, thermal and oxidative reactions further modify flavor precursors [10]. Maillard reactions between reducing sugars and amino acids intensify, generating pyrazines, alcohols, and aldehydes that impart roasted and nutty aromas. Lipid oxidation produces volatile aldehydes and ketones, enhancing aroma complexity. Simultaneously, acids such as acetic acid volatilize, reducing acidity and balancing flavor. Controlled drying conditions, including temperature and airflow, are critical to preserving these reactions while avoiding the degradation of desirable compounds, ensuring optimal flavor development in cocoa beans.

As mentioned above, drying is a step that is considered to greatly influence and contribute to cocoa flavor quality. In cocoa drying, mass and heat transfer phenomena take place simultaneously (especially heat transfer mechanisms by convection, radiation, conduction, and mass transfer mechanisms by diffusion and evaporation), as well as chemical phenomena (non-enzymatic and enzymatic mediated reactions). The drying operation reduces the moisture content of the cocoa seed from 60% to 7% (wb) or less in order to guarantee the microbiological quality and the elimination of volatile acids stored in the seed at the fermentation stage [10]. Furthermore, drying is a critical stage that strongly affects sensory and chemical quality due to biochemical reactions that occur inside the seed, such as the oxidation of phenolic compounds (PC) and the formation of volatile compounds that play a key role in aromatic quality [11].

There are two principal drying types for cocoa: solar and drying under controlled conditions (DUCC). Both exhibit advantages and disadvantages in terms of efficiency, costs, and effects on microbiological, chemical, and sensory traits. In solar drying, the moisture of the cocoa seeds is removed by the direct solar radiation on the product itself with or without natural air circulation [11]. It requires intensive human labor. The processing time depends on the weather conditions (from 3 days in the dry season to 15 days in the rainy season), but the longer periods of time may result in off-flavors in the cocoa derivatives due to mold growth [10]. The small-scale cocoa farmers that produce small quantities generally use sun drying, while large-scale production models apply DUCC with hot air [12]. DUCC involves the use of heated air to reduce the moisture content of cocoa seeds. Despite DUCC saving time, the high drying rates may prevent the completion of the required oxidative reactions of phenolic compounds and the acid diffusion/evaporation process, and this can generate highly acidic and astringent cocoa beans [13].

A study conducted in Ecuador evaluated the impact of three drying techniques on the volatile profile of fine-flavor and bulk cocoa varieties, using HS-SPME-GC-MS to analyze the effects of oven drying (OD), sun drying (SD), and a modified sun drying method using black plastic sheets (SBPD). A total of 64 volatile compounds were identified, with significant modifications observed after drying, highlighting marked differences between cocoa varieties and a strong influence of the interaction between the drying technique and cocoa type, as indicated by simultaneous component ANOVA. Principal component analysis revealed a similarity in the volatile profile of bulk cocoa dried using OD and SD, while fine-flavor cocoa exhibited variations depending on the applied drying method. From an economic perspective, the choice of drying technique can significantly impact cocoa’s market competitiveness, as inadequate drying may compromise its sensory quality and, consequently, its market value. In this context, the SBPD technique, being simpler and more cost-effective, could represent a viable alternative to enhance drying efficiency, minimize economic losses, and improve cocoa valorization in specialized markets, allowing for the production of fine-flavor cocoa with a comparable aromatic quality or bulk cocoa with an improved volatile profile, relative to the traditional SD and OD methods [14].

Furthermore, the study of the biochemical changes and dynamics that key endogenous compounds inside the cocoa seeds undergo during drying is of special interest, as they are directly related to flavor quality traits. For example, chained carboxylic acids (acetic and lactic) are related to the perception of acidity, and phenolic compounds, methylxanthines, peptides, and diketopyrazines with bitterness; in addition, phenolic compounds are linked to the sensation of astringency, while volatiles are linked with the aroma of the final product [15]. In this way, this paper provides an overview of the state-of-the-art of cocoa drying methods, operating variables, and equipment, and analyzes the effects of drying technology on essential chemical compounds related to cocoa flavor quality, such as organic acids, phenolic compounds, methylxanthines, amino acids, sugars, and volatiles.

## 2. Methodology for the Review Article: Phases of Development

Below is a structured and academic methodology for the review article on cocoa drying technology, segmented into distinct phases. This approach ensures a systematic process for data collection, analysis, and presentation [16,17].

### 2.1. Phase 1: Definition of Objectives

The first phase establishes the goals and scope of the review. The primary objective is to comprehensively evaluate the influence of cocoa drying technologies on flavor formation and quality. This includes: (i) analyzing the effects of drying on the sensory, chemical, and functional traits of cocoa; (ii) reviewing methods, variables, and equipment used in drying processes; and (iii) identifying knowledge gaps and research opportunities for advancing cocoa drying technologies.

### 2.2. Phase 2: Search and Collection of the Literature

The literature search was conducted exclusively in Scopus, one of the most recognized databases for its rigorous indexing and comprehensive coverage of high-quality scientific literature [18,19]. Gray literature (technical reports, theses, institutional documents, or preprints) was not included due to the variability in review and validation processes associated with these sources. Given the study’s focus on peer-reviewed literature published in high-impact scientific journals, we prioritized sources with standardized quality criteria to ensure the reliability and reproducibility of the analysis.

For data extraction and analysis, key thematic areas were defined based on patterns identified in the selected studies. These thematic areas were determined through an iterative process that combined the review of keywords, abstract structures, and the categorization of main topics covered in the literature [20]. The information was then systematically classified within these categories using a qualitative analysis, ensuring a coherent and structured organization of findings. This approach enabled a solid interpretation of the collected information, maintaining the transparency and accuracy of the study.

The following search string was applied: TITLE-ABS-KEY (“cocoa drying” OR “cocoa drying technology” OR “cocoa quality” AND (“solar drying” OR “controlled drying” OR “mass transfer” OR “heat transfer” OR “phenolic compounds” OR “volatile compounds”)), establishing the following article inclusion and exclusion criteria:

**Inclusion Criteria:** Peer-reviewed articles published in the last 50 years. Studies that focus on cocoa drying methods, variables, equipment, and their impact on quality. Articles in English.

**Exclusion Criteria:** Non-peer-reviewed articles, patents, and conference abstracts. Studies unrelated to cocoa processing.

Titles and abstracts were reviewed to ensure that the documents were related to the objective of the study; in this initial review, a total of 12 studies that did not contribute significantly to the study were deaccessioned. Subsequently, a review of the complete texts was carried out to ensure that the documents were related and met the criteria for inclusion. In this step, 2 replicated documents were deaccessioned, leaving a final number of 89 documents that were analyzed. The methodology described above has already been applied in several review articles [21,22,23].

### 2.3. Phase 3: Data Extraction, Categorization, and Analysis

Key information was systematically extracted from the selected articles, focusing on critical aspects of cocoa drying technology, including drying methods (solar, controlled, and hybrid approaches), process variables such as temperature, airflow, and humidity, the design and functionality of drying equipment, and the effects of these processes on the phenolic compounds, volatile compounds, and sensory attributes of cocoa. Once extracted, the data were categorized into thematic areas to facilitate analysis. These categories included the fundamental heat and mass transfer phenomena occurring during drying, the biochemical and physical changes that take place within the cocoa beans, the efficiencies and limitations of various drying technologies, and the overall impact of these processes on cocoa quality traits, such as flavor, acidity, and texture. This structured approach ensured a comprehensive and systematic evaluation of the existing literature, allowing for a detailed synthesis of findings relevant to optimizing cocoa drying technology.

## 3. Cocoa Drying: Process, Variables, and Equipment

Drying is one of the most widely used methods for the post-harvest processing of food matrices and is extensively applied in agricultural products, such as grains, fruits, vegetables, wood, and aromatic herbs [24,25]. Drying is considered a unitary operation in which thermal, physicochemical, or mechanical energy is supplied to a food matrix, with the main aim being to reduce moisture. This process involves simultaneous heat and mass transfer phenomena that are related to the Fourier and Fick laws, respectively [26,27]. Therefore, drying allows the shelf life to be extended, which facilitates the transport and storage of food and significantly influences the chemical, microbiological, functional, and sensory quality of the product [28].

In cocoa transformation from seeds to beans, fermentation is considered the core and most studied stage. However, the quality of cocoa is the result of the sum of genetics and agroecological factors and the influence of processing operations, especially drying. Cocoa drying consists of a thermophysical and biochemical process that occurs in the fermented seeds to remove moisture, to prevent spoilage by microorganisms, and to facilitate the development of chemical reactions to favor the sensory quality of the derived products. As can be seen in Figure 1, cocoa drying involves two transfer phenomena: heat and mass transfer that involve mechanisms and physics laws [29].

### 3.1. Heat Transfer

Three heat transfer mechanisms are involved in these phenomena:

#### 3.1.1. Convection

Convection occurs due to a difference in temperature between the airflow and the inside of the seed. The airflow moves over the seed surface (the testa is the seed coat) or the surrounding area. The outflow of water from the inside of the seed is caused by a moisture differential between the inside of the seed and the air. The physical law that explains the heat transfer by convection (Qconv) is Newton’s cooling law [30].

Convection is the dominant heat transfer mechanism in most cocoa drying methods, as it involves the movement of heated air transferring thermal energy to the beans [31]. In tunnel dryers or fluidized bed dryers, hot air circulates and transfers heat to the beans, evaporating surface moisture and facilitating its removal. In solar dryers, convection naturally occurs when solar-heated air rises, creating an airflow that carries moisture away from the beans [25]. To enhance drying efficiency, fans or forced-air circulation systems can be employed to increase airflow velocity and improve uniformity throughout the drying process.

#### 3.1.2. Conduction

Conduction is the second mechanism that allows the transfer of energy from the most energetic seed particles to the least energetic as a result of the interaction between them. In solids such as cacao, heat transfer is due to the combination of lattice vibrations of the molecules and energy transport by free electrons. It is governed by Fourier’s law [32].

Heat conduction in cocoa drying occurs when thermal energy is transferred directly from a heated surface to the cocoa beans in direct contact with it. An example of this mechanism is the use of metal trays in hot-air dryers or drying tunnels, where heat from the tray material is conducted to the cocoa beans, promoting moisture evaporation [33]. In solar dryers with a metal base, beans also absorb heat by conduction from the surface, which is warmed by solar radiation. This process accelerates moisture removal from the bottom layer of beans before heat is more uniformly distributed through convection [26].

#### 3.1.3. Radiation

Energy is emitted by cocoa seeds in the form of electromagnetic waves or photons due to changes in the electronic configurations of atoms or molecules. The rate of the radiation energy per unit area is defined by the Stefan-Boltzmann law [34].

Thermal radiation plays a crucial role in solar dryers, where solar energy is converted into heat, which is absorbed directly by the cocoa beans and drying surfaces [35]. In greenhouse-type solar dryers, transparent materials such as polycarbonate or glass allow solar radiation to enter, increasing the internal temperature and improving drying efficiency. Additionally, heat-absorbing surfaces, such as black-coated drying bases, optimize radiation absorption and enhance heat transfer to the beans, contributing to a more effective moisture reduction process [26].

### 3.2. Mass Transfer

Mass transfer occurs due to a moisture gradient between the desiccant/hot air and the seed. It causes the transport of moisture from inside the seed (where the moisture is higher) to the surface/testa (where it is lower) through diffusion (Figure 1). Finally, the transport of moisture occurs from the seed surface to the surrounding air by convection, and evaporation is the mechanism for the mass transfer (Figure 1). Therefore, as this change of phase occurs, the transfer of mass and heat must be considered simultaneously [36]. In addition to moisture, the evaporation of undesirable compounds occurs, such as short-chained carboxylic acids, including acetic, propionic, butyric, and isobutyric acids.

The diffusion of moisture during drying is a complex process that includes the following processes: molecular diffusion, capillarity of flow, Knudsen flow, hydrodynamic flow, diffusion surface, and flow caused by the sequence of vaporization and condensation, which together explain the moisture diffusivity that is defined by Fick’s law [37]. Thus, the diffusional theory states that the mass flux per unit area is proportional to the water concentration gradient (Equation (1)). Through the mathematical solution of equation 1, it is possible to analyze the drying rate and drying time.(1)$$\frac{\partial X}{\partial t}=\nabla (D_{ef}\;\nabla X)\;\;$$
where ${\;D}_{ef}$ (m^2^s^−1^) is the effective diffusivity, X (Kg_H_2_O_ kg^db−1^) is the moisture content in the product on a dry basis, and t is the time. The most commonly used types of cocoa drying processes are sun drying and drying under controlled conditions (DUCC) with hot air [13]. The phenomena of the process, variables, equipment, and their effects on the cocoa quality will be addressed below.

In cocoa drying, mass transfer is a complex phenomenon influenced by the moisture gradient between the beans and the drying air [38]. Water migration from the interior of the beans to their surface occurs through diffusion, followed by evaporation facilitated by convection. In addition to moisture removal, this process also enables the volatilization of undesirable compounds, such as short-chain carboxylic acids, which can affect the sensory profile of cocoa [39].

In mechanical or forced-air dryers, increasing the temperature and reducing the relative humidity of the air enhance water evaporation from the cocoa surface, allowing internal moisture to progressively migrate outward. In contrast, in solar dryers, mass transfer can be slower if environmental conditions are not optimal, highlighting the importance of proper ventilation control to improve process efficiency [26]. The application of mathematical models, such as Fick’s Law, allows for the analysis of drying kinetics and the optimization of variables such as temperature, airflow velocity, and relative humidity [40], ensuring an efficient drying process without compromising the cocoa’s sensory quality.

### 3.3. Types of Cocoa Drying

#### 3.3.1. Solar Drying

##### Process: Principles and Mechanisms of Heat and Mass Transfer Phenomena

In solar drying, the heat generated by the sun’s rays is used as the sole or partial source of radiation, with or without the natural circulation of air by convection. The principal aim is to reduce the moisture content in cocoa seeds mainly by evaporation and to allow the biochemical changes that are induced by fermentation to be completed so that high-quality sensory traits are obtained. This process can take between 3 and 15 days depending on the climatic conditions [41].

As can be seen in Figure 2, solar drying is due to solar radiation and, at the same time, to natural convection. The incoming solar energy is divided into two; one part is reflected, and the other is absorbed by the fermented cocoa seeds, converting it into thermal energy to heat the seeds. Longwave radiation losses from the cocoa surface to the environment are generated through moist air, and simultaneously, heat losses occur by convection and evaporation due to wind blowing through moist air on the cocoa seed surface. Even conduction heat losses can be generated depending on the drying surface (concrete, wood, and stainless steel, among others) [42].

##### Solar Drying Methods and Equipment

The solar drying process is carried out by placing the fermented cocoa seeds directly under the climatic conditions of the environment: solar thermal energy, temperature, relative moisture, and natural airflows [43,44]. Some cocoa farmers spread fermented cocoa onto the drying surfaces, forming small cocoa piles to a depth of not less than 5 cm during the first hours of drying (Figure 3A), to avoid violent drying and the consequent crusting of the testa, which leads to the obtention of acidic cocoa that negatively influences the flavor quality. On the second day of drying, farmers spread out and mix the cocoa beans over the entire surface of the drying equipment [45]. The practice is dependent on weather conditions that are not always favorable for this type of drying. In the rainy season, cocoa producers must take advantage of daylight hours from the first day, since the environmental conditions (rain and high humidity) prolong drying, and cocoa can be infected with fungal pathogens.

Solar drying involves the use of large surfaces and long process times, reported to be between 3, 7, and 15 days, and even up to 22 days in the wet season, in order to obtain the ideal final moisture content for cocoa that is between 6 and 7% (wb) [13]. During drying, the bed of cocoa beans placed as a monolayer on the drying equipment is mixed to change the position of the totality of the beans and to rotate the side that is exposed to the natural air stream. The mixing process is carried out manually with wooden rakes, and the frequency (i.e., twice a day) depends on the farmer’s time constraints. Mixing ensures uniform drying and breaks up the cocoa agglomerates. This means that solar drying developed on small farms is very labor-intensive and has low operational performance [12].

The drying rate is critical for the flavor quality of the cocoa beans. When the drying rate is too slow, particularly in humid or rainy conditions, where it can take up to 22 days to reach the desired moisture content (6–8%) [13], the possibility of growth of fungal pathogens increases, which penetrate the testa and develop inside the bean, creating a product with unpleasant flavors (i.e., earthy, spicy, moldy, putrid, hammy, rancid, etc.). On the contrary, when the drying rate is too rapid, cocoa beans can reach the desired moisture content of 6–8% within 18 h [13]; the crusting phenomenon of the testa occurs, preventing the mass transfer of acids to the outside of the seed, which causes excessive acidity. The premature hardening of the testa also prevents the entry of oxygen flow. Thus, oxidative reactions on phenolic compounds may be interrupted, generating an increase in bitterness and astringency in cocoa [46].

In general, previous studies have reported that solar drying produces cocoa with higher sensory quality and less non-specific and undesirable aromas, such as rubbery, smoky, or gasoline-like, than the cocoa generated by controlled drying [47]. Even though controlled drying is becoming more popular, the sun-drying method is widely used by small-scale cocoa farmers.

Despite the advantages of solar drying as regards sensory quality and the fact that it is environmentally friendly because no electrical energy is necessary, this technique is highly dependent on weather conditions and does not allow control of the operating variables, such as temperature, airflow, and time of drying for optimization. Therefore, the quality of cocoa batches is heterogeneous. Additionally, when the drying method is direct, there is a risk of contamination from the environment, especially due to infestation of insects and birds, the growth of microorganisms, and the development of mycotoxins [48,49]. Consequently, the management of drying conditions and the production of cocoa with homogeneous quality characteristics is a challenge.

The equipment for solar drying differs in terms of the design, materials, and the mechanism used to protect the cocoa from environmental factors that can negatively affect the quality [45]. Moreover, indirect solar drying generally produces cocoa beans with higher resistance properties, with lower levels of titratable acidity and a darker color compared to direct drying [50]. The most employed solar drying equipment is mentioned below:

**Sliding surface or sliding roof dryers:** The drying area consists of a wooden or concrete platform that is above the ground, covered by a mobile roof that is mounted on rails. In the case of sliding apparatus, it is common to use wooden platforms that slide across rails and can be kept indoors depending on weather conditions (in case of rain). Generally, this kind of equipment is called *elbas*, *barçasa*, or *boucan* [51]. There are more sophisticated designs that use multiple platforms arranged one on top of the other to use the available area more efficiently (Figure 3B).

**Solar dryers using mats:** They are located approximately 1 m from the ground and are made from bamboo and dry cane (*Gynerium sagittatum*). These have spaces between the salts that allow the air to circulate. This kind of platform facilitates drying because the drying surface dries easily with solar radiation (Figure 3C). However, it does not usually have a roof to cover the cocoa in case of bad weather conditions [45].

**Greenhouse-like dryers:** Fermented cocoa seeds are placed in an enclosure, with plastic transparent covers or side panels (Figure 3D). Heat is generated by the absorption of solar radiation on the fermented cocoa seeds and by the internal surfaces of the drying chamber. The heat generates the evaporation of the moisture, and there is a natural circulation of the air. They have platforms made of wood or plastic nets and a plastic roof. The main problem with this kind of equipment is the lack of ventilation. It is suggested that designs consider removable side walls and ventilation ducts in the roof through which the moisture can escape [52].

To solve this problem, Sianipar [53] designed and constructed a prototype of a solar-powered greenhouse-like dryer for rural areas in Indonesia based on a participatory diagnosis with cocoa stakeholders. The dryer used locally available materials. In general, the components of the dryer are wooden, except for the drying mat (flat zinc plate), the solar light collector/heat generator in the air heater (corrugated zinc plate), the heat collector, and the transparent covers (UV plastic). It could easily be constructed by small-scale cacao farmers. Validation in the field demonstrates that the cocoa dryer is environmentally friendly and functions properly in a rural context that lacks economic resources and know-how. The dryer reaches an average temperature of 57 °C, and for mainly sunny days, the drying time is two days, while for mainly cloudy or gloomy days, three days are required. In addition, the cocoa beans produced meet the quality standards demanded by markets (level humidity below 7% wb).

A variation of greenhouse-like dryers consists of chamber dryers (tray and tunnel type) and chimney dryers. Fermented cocoa seeds are spread out in a chamber with a plastic or glass cover, which is ventilated through a series of holes. Heat transfer is indirect when solar energy is absorbed into a collector and directed into the chamber. These dryers depend on airflows to extract moisture from the dehydration chamber, although fans can be incorporated to circulate the airflow. In this way, Simo-Tagne et al. [33] evaluated a solar dryer with a drying chamber, a solar collector, a fan, and a chimney to remove the cocoa bean moisture. The authors developed a simulation model for natural, forced, and combined natural-forced convection solar drying of cocoa beans under natural environmental conditions using the thermo-physical properties of cocoa beans. The most promising results in terms of the time and efficiency of the drying process were obtained when the combination of methods, forced convection during sunny periods (day) and natural convection during the night, was applied. As expected, the drying time was longer for natural convection, while 32 h were required to obtain a final moisture content of 0.15 kg/kg (db) using the combined method. However, the desired moisture required by the market was 0.08 kg/kg. The natural convection method exhibited lower global thermal efficiencies located between 5 and 8%, whereas the forced and combined methods showed efficiencies between 8 and 17%.

Additionally, hybrid solar dryers were studied that were designed to store energy during the active drying period and release it at night during the resting period so that the drying process could continue. This implies that at night, when the ambient temperature decreases and the relative humidity increases, cocoa beans do not reabsorb moisture from the environment, leading to a shorter drying time and a high-quality product. In this way, Adeyemi et al. [54] investigated the drying kinetics of cocoa using a mixed-mode solar drying method. The drying was divided into two phases: the active drying period (day) and the resting period (night). The intermittent drying curve showed a constant drying rate and a falling drying rate. Moisture loss was rapid during the first six hours of the first drying day and continued at a lower rate for the next six hours, extending to the resting period. The rapid moisture loss during the first few hours of the first day was attributed to the interstitial moisture migration of the free moisture at the surface of the fermented beans. The total drying time was 61.5 h. The results showed that this mixed-mode solar dryer is a promising alternative for drying cocoa, as the process is quicker, preserves cocoa quality, and is not influenced by adverse weather conditions.

**Drying surfaces:** Some small-scale cocoa farmers use cane mats (Figure 3C), cement (Figure 3E), or plastic nets as surfaces to spread fermented cocoa seeds. Even metal surfaces, such as zinc or the edges of asphalt roads, are used (Figure 3F). The latter negatively affects the final cocoa quality, as smoke is generated that may contain polycyclic aromatic hydrocarbons, which are considered carcinogenic [45]. Additionally, simple technologies such as windbreaks could help regulate airflow and minimize disruptions caused by strong winds, ensuring more consistent drying conditions. Elevated platforms, on the other hand, could reduce contamination risks from soil or debris while improving air circulation around the beans. Combining these techniques with better mixing practices during drying to prevent uneven moisture loss would provide further benefits. This would help bridge the gap between traditional methods and the need for more consistent, efficient practices, ultimately benefiting small-scale farmers and improving cocoa quality.

Finally, it is crucial to analyze the specific characteristics of each solar drying method and its impact on process efficiency and cocoa quality. Table 1 presents the main types of solar dryers used, detailing their designs, advantages, and limitations. This information provides a comparative assessment of the available alternatives, facilitating informed decision-making for the implementation of more efficient drying techniques adapted to the specific conditions of each region.

#### 3.3.2. Drying Under Controlled Conditions (DUCC)

##### Mechanisms of Heat and Mass Transfer Phenomena and Methods and Equipment for DUCC

DUCC involves the removal of moisture content from the fermented cocoa seeds using a flow of hot air by forced convection. The operating variables, such as air temperature, moisture, and flow, are controlled by the equipment [24]. The surrounding hot airflow that is in contact with the surface of the seed causes an increase in the temperature of the seed. The hot airflow is the medium used to supply the energy that evaporates free water from the seed and removes it to the surrounding atmosphere, generating the mass transfer (Figure 4) [30]. In this way, as the surface moisture evaporates, the transport of moisture from the inside to the surface of the seed occurs mainly through one or more of the following mechanisms based on Fick’s law of diffusion: liquid diffusion (if the seed temperature is below the boiling point of water), vapor diffusion (if water evaporates within the seed), and hydrostatic pressure differences due to internal stress causing shrinkage of the seed [37]. This fundamentally depends on the external conditions of the temperature (*T_a_*), moisture (*X*_a_), rate (*V_a_*), and direction of the airflow, as well as the properties and geometry of the seed [55]. Two types of DUCC can be developed at constant conditions (air temperature and velocity) and unsteady conditions, using intermittency cycles in the process variables with rest cycles during the process.

Additionally, different heat sources can be used in DUCC, those in direct and indirect contact with the fermented cocoa seeds to remove moisture content. The heat source used for direct drying can be air heated by water vapor or solar radiation, combustion products, inert gases, or superheated steam. When a direct heat source is used, such as wood fires, an unpleasant flavor characterized by smoky and hammy aromatic notes is perceived in the cocoa beans. This is in concordance with the results of Streule et al. [56]. The authors reported that high temperatures were reached at certain points during direct convective drying depending on the equipment used; thus, temperatures of up to 110 °C were reached. Regarding these high temperatures, when a small gas-powered direct dryer with close contact to the fire without ventilation was used, the severe temperatures resulted in a roasted-burnt off-flavor when the dried beans were sensory evaluated. Because of these negative influences on flavor, indirect sources are mostly used for DUCC. The heat sources for indirect drying can be condensed steam, hot water, thermal oils, combustion gases, or electrical resistance [57].

In general, the research focused on using DUCC for cocoa, analyzing the drying kinetics, the degradation kinetics of phenolic compounds, the modeling and simulation of drying, and the effect of drying variables on the concentration of acids, methylxanthines, and volatiles, as well as on some sensory attributes. Furthermore, prototypes for DUCC have been designed and constructed to assess the energy efficiency and to analyze the temperature distribution and moisture in cocoa. Finally, a combination of drying methods (convective, adsorption, vacuum drying, and freeze-drying), as well as the application of intermittence during convective drying, was considered as alternative methods to preserve the functional potential of cocoa and to avoid excess acidity due to the crusting phenomena that negatively influence the flavor quality.

The research related to the methods and equipment used in DUCC is mentioned in the following sections.

##### DUCC Equipment: Hot-Air Oven, Flatbed, Rotary/Drum, Tunnel, Fluidized Bed, and Heat Pump Driers

*Hot-Air Oven Drying:* a common drying system that uses convection to circulate hot air around a product. The heated air, generated by electrical elements or gas burners, removes moisture by evaporation. Research that assessed the forced convective drying of cocoa with hot air focused on evaluating the reduction of process time to make it efficient in terms of energy saving, and beyond this, the design and evaluation of automated convective drying equipment were considered. Generally, the drying rate rises when the temperature increases, but this has negative effects on the sensory profile of the final product, due to the interruption of two processes: the oxidation reactions of phenolic compounds and the evaporation of volatile acids, such as acetic acid, as a result of the crusting [13].

To solve this problem, several researchers have studied applying constant drying and using *convective drying ovens* at temperatures from 30 to 80 °C and air velocities between 0.3 and 4 m/s [38,58,59,60,61,62,63,64,65,66]. Briefly, this previous research evaluated the drying kinetics, employing theoretical and semi-theoretical mathematical models, the degradation kinetics of phenolic compounds (total polyphenol content, flavan-3-ol and procyanidin concentrations), and, in a few cases, the sensory quality traits of chocolate produced from dried beans, such as acidity, cacao taste, and astringency, were evaluated. The main findings highlighted that the drying air temperature has a strong influence on the drying kinetics, while air velocity has a small but not negligible influence on the drying rate. The drying rate increased, with an increase in temperature and air velocity, but decreased over time. On the other hand, drying at lower temperatures, below 60 °C, and when the rate curve in the first drying stage is constant, leads to a reduction in drying time due to the constant loss of moisture. On the other hand, this drying method has a severe negative effect on the concentration of phenolic compounds, attributed to the higher temperature profile incurred during drying. Thus, catechins and flavonoids are the most degraded, and an intense reduction of these bioactive compounds is observed.

Furthermore, it was stated that temperatures lower than 60 °C favor the cocoa sensory profile while drying at high temperatures (80 °C), with direct heat contact resulting in beans with a roasted to burnt off-flavor [38,58,59,60,61,62,63,64,65,66]. These results should be interpreted carefully to ensure good sensory cocoa quality, as it is necessary, in addition to kinetic and physicochemical analyses, to carry out sensory analyses (preferably descriptive) in cocoa derivatives, such as chocolate bars derived from this type of dried bean. This is to evaluate the effect of drying conditions on the perception of flavor attributes.

Additionally, it was observed that the lower the final moisture content of the hot air-dried samples, the lower the percent retention in phenolic compounds. Moreover, cocoa beans processed by convective oven dryers tend to have high levels of acidity and lower concentrations of phenolic compounds compared to solar drying. For this reason, as an alternative, convective drying in an oven of the fermented seeds at lower temperatures for a longer time has been proposed. However, this method has disadvantages, as the phenolic concentration in the cocoa beans is considerably lower due to intense enzymatic degradation over the processing time when longer drying times are applied.

An advantage of drying with automated convective dryers is the possibility of using Internet of Things technology, which stands out as an innovative tool that allows remote access to information, drying process management, and the control of variables. Furtado et al. [67] developed a low-cost automated convective dryer for cocoa coupled to an application for smartphones. It allowed real-time activation via an internet connection, receiving data, recording data from the sensors, and triggering the dryer fan in real time. The captured and stored data exhibited consistency according to the sensors during drying. Furthermore, the equipment allowed data analysis through the cloud database, providing parameters (moisture and temperature) for decision-making remotely.

Furthermore, substitute equipment includes convection flatbed dryers. Flatbed dryers consist of a perforated bed where products, such as grains or seeds, are spread in a thin layer. Hot air is forced through the bed, drying the product evenly. These dryers are widely used in agricultural sectors due to their simple design and ability to handle large batches. However, they require the regular turning of the product for uniform drying. Using this equipment, Tardzenyuy et al. [68] showed the advantages of using a prototype flatbed dryer on dry fermented cocoa beans. The authors stated that, at a temperature of 35 °C, an aeration rate of 15 m^3^/s, a space/quantity ratio of 12 m^2^/50 kg, a process time of 96 h, and a final bean moisture of 7.0% (wb) is reached, despite the weather conditions. Similarly, Komolafe et al. [69] focused on the evaluation of a wood flatbed dryer with a capacity of 25 kg batches. The moisture content was reduced from 80.01% (wb) to 7.49% (wb) after 7 h of continuous drying with temperatures between 61 °C and 67 °C. The drying efficiencies were between 72.3% and 92.9%, respectively. However, none of these studies analyzed the effects of these drying methods on the bean sensory quality.

On the other hand, some studies have evaluated other equipment, such as drum/rotary, tunnel, fluidized bed, and heat pump dryers with the potential to improve thermal efficiency, the quality of cocoa, and how the equipment could be adjusted to the production requirements of smallholders. Castellanos et al. [70] investigated the concentration of phenolic compounds and methylxanthines in dried cocoa beans (at 50 and 60 °C) using a rotary dryer equipped with an electric heater and a set of infrared lamps. Rotary dryers feature a rotating drum that tumbles the material while hot airflows through it. Their efficiency lies in consistent heat transfer and drying speed. The results indicated that there are no significant differences in the content of total polyphenols when the temperature increases. However, the concentration of (+)-catechin (from 6.8 to 2.6 mg/g), (-)-epicatechin (from 3.8 to 2.2 mg/g), theobromine (from 22 to 13 mg/g), and caffeine (from 1.4 to 0.6 mg/g) decreased when a higher temperature was applied. Furthermore, when the drum dryer coupled to infrared radiation was used, the results related to the content of phenolic compounds and methylxanthines are comparable with those obtained in fresh cocoa and solar-dried samples. Despite these findings, other quality criteria related to organoleptic traits and color assessment need to be addressed to have a more comprehensive study of operating variables that affect the product.

Another mechanical dryer that can improve the global quality of cocoa beans, allowing control of the process and reducing drying times, is the tunnel dryer. Tunnel dryers involve a conveyor or tray system where products move through a tunnel as hot air circulates. Alean et al. [71] reported that, with an air temperature of 40 °C and a velocity of 3 m/s, drying can be achieved over 24 h, obtaining a high concentration of polyphenols at the end of the process, with 3329.76 mg gallic acid/100 g db. Whereas the authors observed that the highest degradation occurred at 60 °C.

A laboratory-scale prototype of a heat pump for cocoa drying was evaluated by Hii et al. [47,72], who investigated the drying kinetics of cocoa beans using this equipment and simulated heat and mass transfer in 3D. A heat pump dryer uses a closed-loop system to extract moisture from materials efficiently. It operates by dehumidifying the air in the drying chamber using a heat pump, which transfers heat from the surroundings or exhaust air to the drying process. This system is energy-efficient as it recycles heat, reducing energy consumption compared to traditional dryers. Heat pump dryers are ideal for temperature-sensitive products, such as cocoa beans, as they maintain a controlled, low-temperature environment to preserve quality, aroma, and nutrients while ensuring thorough drying. The heat pump dries the fermented cocoa seeds under dehumidified conditions and recovers the latent and sensible heat of water evaporation. The main components of the prototype are the heat pump, the heat exchanger, and the drying loops. This kind of dryer is based on the refrigeration cycle and heat exchanges with the drying air that moves through the condenser (air heating) and evaporator (air cooling); however, there is no air exchange between the drying air and the surrounding air. Simulation results showed that the change rate of the temperature within the seeds was significantly faster compared to the reduction in moisture content during drying, while the shrinkage was negligible. In addition, the predicted and experimental data related to bean temperature profiles exhibited a good fit between them, with a mean relative error of less than 5%. Finally, Hii et al. [47,72] emphasized that heat pump dryers can improve cocoa quality traits, such as color and texture, and beyond this, they can preserve thermolabile bioactive compounds, particularly phenolic compounds. For example, Hii et al. [72] reported higher retention levels of cocoa polyphenols in the range of 44% to 73% compared to freeze-drying, while cocoa bean hardness was comparable to the commercial cocoa samples and increased when moisture content decreased.

Another piece of drying equipment that leads to a rapid and economic drying process is the swirling fluidized bed dryer. A swirling fluidized bed dryer is an advanced drying system where hot air is passed through a perforated bed containing the material to be dried, causing it to become fluidized. This creates a dynamic interaction between the air and the particles, resulting in rapid and uniform drying. The high heat and mass transfer rates make it highly efficient for materials like grains, powders, or granules. It allows for the thorough mixing of beans, and it ensures efficient mass and heat transfer. It was studied by Zulkarnain et al. [73]. When this tool was compared to solar drying equipment, the authors observed that the fluidized bed dryer was more efficient as the drying time lasted two days at 35 °C, the weight loss reduction was higher (59.16%), and the moisture ratio was lower (0.4083) compared with solar drying, which took 11 days with a weight reduction of 58.55% and a 0.4145 moisture ratio. Despite this, analyses of physicochemical and sensory properties were lacking in terms of guaranteeing an integral study, not only including operational process parameters but also cocoa bean quality.

Finally, other studies have focused on analyzing the influence of combined solar and convective drying methods on physicochemical parameters, such as pH, acidity, and free fatty acid content. Zahouli et al. [10] and Guehi et al. [12] applied a combined drying method that consisted of solar drying from 9 h to 18 h daily to attain a moisture content of 25% (wb) and then an artificial drying process using an air-ventilated oven at 60 °C to attain a moisture content of 7% (wb). For the oven drying, the process was conducted for 8 h daily, and the beans were tempered at room temperature overnight. The principal findings show that the mixed drying method was comparable with the solar drying method. The pH of solar-dried beans was 4.5 to 5.5, and for mixed dried beans it was between 3.8 and 5.2. In this way, both produced less acidic cocoa beans because the slow and moderate drying process enables the evaporation of more acetic acid quantities. The authors also found that solar-dried beans and mixed-dried beans produced low free acidity when compared to oven-dried beans. The only point of difference between the two methods was in the volatile acidity. Solar drying showed lower volatile acidity than the combined method. As observed, the findings of the combined drying methods are promising and contribute to obtaining cocoa with acceptable physicochemical quality characteristics.

In addition to convective drying, other drying technologies such as microwaves, ultrasound, and infrared radiation can also be considered to define hybrid methods that could contribute to the preservation of bioactive compounds [74,75]. However, the application of advanced non-conventional drying technology for cocoa is not currently used a great deal [63]. Recently, Alean et al. [36] applied microwaves to investigate the in situ oxidation of polyphenols during drying. In this study, the authors also reported that the intermittent ON-OFF technique using microwaves, specifically an ON period of 5 s, allows for a significantly higher percentage of polyphenols (86%) to be retained than the solar drying method (60%). Despite this, research on the application of these emerging technologies in cocoa is scarce. For this reason, future research on drying could be carried out to evaluate the operating variables of the hybrid/combined methods, to study their effects not only on physicochemical parameters, but also, as previously highlighted, to consider their influence on the organoleptic and functional characteristics of the cocoa beans and process efficiency.

The impact of dryers has been mostly analyzed through physicochemical parameters, and descriptions of their effect on the sensorial attributes of the final chocolate are scarce. Nonetheless, this is the ultimate objective, and this review evidences the lack of data on this topic.

##### Intermittent Drying Method as an Alternative for Preserving Cocoa Quality

DUCC using a convective mechanism has several limitations. Cocoa quality is not homogeneous due to excessive violent drying, which is when the conditions are extreme, with high drying rates or temperatures, and the hardening of the testa takes place, reducing the diffusion/evaporation of volatile acids, resulting in acidic cocoa beans. Furthermore, the enzymatic hydrolysis of anthocyanins stops, and purple and astringent seeds are obtained [76]. This implies that operational variables of convective drying should be controlled to avoid low yield, low efficiency, high operating costs, and negative effects on the quality and food safety of cocoa beans [28]. Research has sought to develop alternatives to reduce these limitations, and, additionally, studies have focused on studying the degradation kinetics of phenolic compounds, the antioxidant capacity, and ochratoxin A [63,77,78]. For this reason, cocoa drying should be carried out slowly and under controlled conditions to preserve the phenolic compounds and to obtain high-quality flavor profiles in cocoa derivatives [71]. Thus, new alternatives to conventional drying methods under constant conditions have been developed. The application of intermittent drying under controlled conditions of fermented cocoa seeds seemed a promising option.

Controlled intermittent drying is based on the combination of operating conditions that vary during the process, such as heat supply, temperature, and drying air velocity [79]. This method of drying could improve the functional and sensory quality and reduce the problems related to crusting since it decreases the surface temperature in the seed, the exposure time to the heat treatment, and the heat necessary to evaporate the surface water. Consequently, an increase in the energy efficiency of the process is produced [80].

The use of intermittent drying with tempering periods generates a moisture gradient inside the seed that allows the migration of moisture from the interior to the surface, causing the moisture throughout the fermented cocoa seed to be almost uniform. This new distribution of moisture facilitates drying, reduces water and temperature gradients, and therefore reduces thermal and water stress, and prevents physical damage (i.e., cracking and hardening of the testa), allowing the diffusion/evaporation of volatile acids to the outside and reducing non-enzymatic browning [81]. Additionally, in terms of efficiency, intermittent drying may reduce the effective drying time.

Research focusing on the application of intermittent drying in cocoa is still limited. A study carried out by Faborode et al. [82] gave promising results after applying intermittent drying to preserve quality. The authors applied a method based on applying forced air at 60 °C at a rate of 0.2 m/s, employing resting periods of one and two cycles. They found that the intermittent periods during drying may result in good cocoa quality, but more research is required to specify the resting sequence to improve the quality. Both the cocoa liquor derived from the fermented cocoa seeds drying under two rest periods of 8 and 28.5 h and the liquor from the continuous drying at 40 °C presented a high score in global sensory quality. The liquor derived from cocoa processed under intermittent drying was more astringent and bitter, and the cocoa taste was slightly weaker.

Intermittent drying is an understudied alternative with high potential to enhance chemical, functional, and sensory quality in cocoa and reduce the effective drying time. Future research should focus on evaluating the optimal combination of the process variables, including resting periods, air velocity, and temperature, to maximize the preservation of bioactive thermolabile compounds in cocoa beans, such as flavonoids and volatile aromas. The above, considering the needs of niche markets to produce healthier and finely flavored chocolate bars, offers economic advantages regarding cocoa prices.

Finally, drying under controlled conditions (DUCC) utilizes various types of equipment to optimize thermal efficiency, reduce drying time, and minimize the impact on cocoa quality. Each drying system has distinct advantages and disadvantages in terms of performance, energy consumption, and its effects on the chemical and sensory properties of cocoa beans. Table 2 provides a comparative overview of DUCC dryers, summarizing their design, efficiency, and potential impact on cocoa quality.

### 3.4. Modeling in Cocoa Drying

Mathematical modeling plays a vital role in optimizing the cocoa drying process by providing insights into how to reduce drying time, increase energy efficiency, lower costs, and enhance the quality of the final product [20]. The cocoa drying process can be compared to drying clothes in the sun. When we hang wet clothes, the air temperature influences how quickly they dry, just as it does with cocoa beans. If there is wind, the airflow helps remove moisture, accelerating the process, while on humid days, clothes take longer to dry because the air already contains a lot of water—the same occurs with cocoa in humid environments. Additionally, if clothes are not left to dry long enough, they may remain damp and develop unpleasant odors, just as inadequate drying in cocoa can affect the quality of the beans. On the other hand, excessive heat can harden or shrink clothes, just as overly aggressive drying in cocoa can degrade key compounds like flavonoids and phenolic acids, impacting its flavor and quality. Therefore, drying cocoa is not just about removing water but finding the right balance between temperature, airflow, and time to preserve its quality. The primary goal of these mathematical models is to characterize and predict the drying kinetics, which refers to the rate at which moisture is removed from the beans. This involves analyzing how water concentration behaves at different temperatures and relative humidity levels [71]. For instance, when cocoa beans are exposed to hot air, the model can help predict how quickly moisture will evaporate based on initial moisture content, air temperature, and airflow rate.

Mathematical models incorporate complex equations governing heat and mass transfer, represented as a system of partial differential equations. These equations account for various factors, such as the physical properties of the cocoa beans, boundary conditions, and the interaction between the beans and the drying air. To solve these equations, researchers use numerical or analytical methods, allowing them to simulate and validate the model against experimental data [73].

Through empirical, semi-theoretical, and theoretical approaches, previous research has extensively analyzed drying kinetics, considering the impact of process variables and the geometry of cocoa beans. Notably, models assess the loss of key bioactive compounds, such as flavonoids and phenolic acids, throughout the drying process, thus capturing the effects on the physical, chemical, and sensory quality of the beans [71,72].

By developing these models, drying conditions can be standardized, allowing for the design and adjustment of drying equipment to optimize essential drying variables. Table 3 gives an overview of research focused on mathematical modeling in cocoa drying. Empirical, semi-theoretical, and theoretical models were proposed to analyze the behavior of the drying kinetics under the influence of process variables, as well as the geometry of the bean and the interaction between the testa, the bean, and the environment. Research for both solar drying and drying under controlled conditions also assessed modeling considering the loss of water, acetic acid, and phenolic compounds over time, the analysis of the effect on the physical, chemical, and sensory quality, and the conservation/retention of compounds with functional potential, such as flavonoids. For that reason, these models are key mathematical tools to standardize the process conditions in terms of air velocity, temperature, and moisture content, and in doing so, design and adjust the drying equipment to optimize and control drying variables.

### 3.5. Effect of the Drying Technology on Key Flavor Compounds Involved in Cocoa Quality

#### 3.5.1. Phenolic Compounds (PC): Flavonoids

The quantity of phenolic compounds in cocoa beans can vary between 5% and 20% (*w*/*w*). It depends on the cultivar, origin, agricultural practices, agroecological conditions, postharvest stages (preconditioning, fermentation, and drying), and processing operations of chocolate (roasting and conching). Cocoa has three main classes of flavonoids. First, procyanidins, or condensed tannins (approximately 58% to 65% of the total polyphenols), are mostly flavan-3,4-diols, which are 4 → 8 or 4 → 6 bound to dimers, trimers, or condensed oligomers based on the epicatechin subunit. Among the procyanidins, oligomers such as B1, B2, B5, and C1 stand out, although procyanidins with a degree of polymerization up to a decamer have been identified [15]. Second, monomeric flavanols (or flavan-3-ols) represent 37% of the total phenolic compounds. From these, (-)-epicatechin is the most abundant, and in lower quantities are (+)-catechin, (+)-gallocatechin, and (-)-epigallocatechin [89]. Finally, anthocyanins are in approximately 4% of the total polyphenols. They are water-soluble pigments responsible for the color of cocoa beans and derivatives. The anthocyanins identified in the cocoa bean include cyanidin derivatives, such as -3-α-ʟ-arabinosid and -3-β-ᴅ-galactosid [15].

PCs are biomolecules of interest for their functional properties, such as high antioxidant activity and bioactivities in industrial, cosmetic, and health-promoting applications, and are responsible for the sensory perception of astringency, bitterness, and color in cocoa derivatives [90]. These sensory attributes significantly influence the balance of the flavor profile that is critical to defining cocoa as a fine or a bulk product [91]. To gain more knowledge on this, the relation between PC and the astringency perception was studied. Ziegleder, in Ref. [92], stated that the presence of soluble flavanols, such as epicatechin and the smallest procyanidins of up to three subunits, cause the astringent taste sensation of cocoa, due to interactions and precipitation with salivary proteins, and, on the contrary, large molecules with more than three subunits are insoluble and are not involved in this sensory attribute. In contrast, Oracz et al. [93] stated that both the insoluble and soluble tannins are responsible for the typical astringency and bitterness of fresh cocoa seeds due to the ability of tannins to form complexes with salivary proteins. These findings are in line with Fayeulle [15], who considered that low-molecular-weight flavan-3-ols (notably epicatechin) are responsible for the bitterness and are also related to marked astringency sensations, while those of higher molecular weight have been described as astringent, and this sensation increases with chain length. On the contrary, it was determined that anthocyanins do not cause bitterness or astringency. These results were in concordance with Luna et al. [94], who associated the phenolic compounds with key sensory traits of cocoa to obtain possible associations between chemical and sensory data. It was found that polyphenols were positively correlated to astringency, bitterness, and the green note, and negatively correlated with fruity aromatic notes. It was observed that with a higher level of polyphenols, the fruity aroma was less perceived in the cocoa liquor due to bitterness and astringency masking the fine cocoa flavor.

Different approaches may arise from variations in the methodologies, experimental conditions, or the types of flavonoids studied in the referenced works. This could involve analyzing the structural features of flavonoids, such as the degree of polymerization, hydroxylation patterns, and molecular size, in relation to their interactions with salivary proteins, which are known to drive astringency perception [95]. Considering that the perception of astringency is primarily driven by the ability of flavonoids to interact with salivary proteins, which is determined by their molecular size, degree of polymerization, hydroxylation patterns, and structural flexibility. While environmental and matrix factors can modulate this interaction, the fundamental driver is the binding affinity and precipitation potential of protein-flavonoid complexes.

#### 3.5.2. Effects of Physical and Biochemical Process Phenomena on the Phenolic Compounds During Drying

The phenolic compounds during drying can be especially affected by the air temperature, airflow, and process time. The drying temperature plays a key role in the retention of phenolic compounds. When the temperature increases, the content of phenolic compounds decreases due to their thermolabile nature. Furthermore, at high temperatures and long drying times, degradation is due to cell destruction [36,96].

Bean moisture is another key variable due to the interaction between phenolic compounds and water. The polarity of phenolic compounds allows them to dissolve and migrate to the surface [36]. During drying, the mass transfer of PCs occurs through molecular diffusion under the influence of vapor partial pressure that diffuses from the cocoa surface into the moving air drying. The PC diffusion from the surface of cocoa beans occurs with the evaporating water, which is controlled by the drying rate [96]. Contrary to this, Alean et al. [71] studied the polyphenol oxidation process in situ during the drying of cocoa beans using microwaves. They reported that the degradation of polyphenol compounds during drying does not obey a transport mechanism, such as diffusion, as had always been stated. In contrast, it is due to in situ oxidative processes. The authors proved that the in situ oxidation of polyphenols is generated as the water evaporates, followed by the penetration of air oxygen to the bean that oxidizes the polyphenols.

##### Chemical Phenomena: Degradation of Phenolic Compounds by Enzymatic and Non-Enzymatic Reactions

***Enzymatic reactions:*** As can be seen in Figure 5, the degradation of PC during drying is mainly due to enzymatic and non-enzymatic reactions. Flava-3-ols monomers are sensitive to biochemical transformations, such as oxidation, polymerization, epimerization, and cleavage [97]. In general, enzymatic degradation occurs under drying at mild/low temperature and water activity, and chemical transformation (epimerization and cleavage) under high process temperature. Oxidation can occur through three main mechanisms: i. direct oxidation by polyphenol oxidases (PPOs); ii. coupled oxidation by transfer from other polyphenols, themselves substrates of PPO, or peroxidases (PODs); iii. autooxidation. Therefore, two enzymes are highlighted: POD, which is an oxidoreductase that uses H_2_O_2_ as a co-substrate to catalyze reactions during drying, and the polyphenol oxidases, with a pH range of optimal activity between 4 and 7 and a temperature between 30 and 50 °C [8]. PPO catalyzes the transformation of phenolic compounds, first into quinones that undergo further condensation with free amine and sulfhydryl groups, leading to the synthesis of brown polymers (melanin) [98]. Kyi et al. [58] studied the kinetics of the polyphenol oxidation reaction in cocoa beans during convective drying and reported that, during cocoa drying, the PPO enzyme is intensively active and can degrade phenolic compounds during the first 5 h of drying. Thus, (-)-epicatechin and flavan-3,4-diols are enzymatically oxidized, producing the flavonoids that give the characteristic brown color to cocoa beans. The authors observed that rates of polyphenol oxidation and condensation exhibited first-order reaction kinetics and are also linearly related to each other. Moreover, PPO is strongly inactivated during drying due to thermal sensitivity, leaving only 2% after drying [99].

On the other hand, the anthocyanins could be hydrolyzed enzymatically by adding glycosidases to anthocyanidins. The latest compounds polymerize together with simple catechins to form insoluble complex tannins of high molecular weight, which leads to a significant reduction in the simple catechin concentration. Therefore, this implies a reduction in the bitterness and astringency of the final product to acceptable levels [100].

***Non-enzymatic reactions:*** The decrease in the phenolic compounds is also attributed to non-enzymatic reactions, such as polymerization and condensation, due to the relatively high temperatures used in the drying process [96]. Furthermore, Hansen et al. [101] stated that the non-enzymatic oxidation of phenolic compounds takes place during the drying process. The autoxidation of flavan-3-ol monomers, dimers, and polymers results in the formation of intramolecular and intermolecular new covalent bonds, while, in proanthocyanidins, terminal-to-terminal unit intermolecular reactions produce linear polymers [89].

During drying, phenolic compounds are hydrolyzed, oxidized, and condensed. Oxidation and condensation reactions, such as the oxidation of protein-polyphenol complexes and the condensation with amino/carbonyl/sulfhydryl groups, contribute to reducing astringency and bitterness perceptions and influencing the flavor and color of cocoa positively. The chemical reactivity with other macromolecules, such as carbohydrates, proteins, and other polyphenols, begins with a small quantity of flavanols bonded to proteins or polysaccharides to produce soluble complexes. This is due to the capacity of phenolic compounds to form non-covalent or covalent bonds. The non-covalent interactions are related to hydrogen bonding, hydrophobic interactions, or π-π stacking. Additionally, two more stages follow. In the second one, if the concentration of flavanols is high enough, aggregates are generated, and finally higher flavanol concentrations lead to the formation of precipitates [89].

Proanthocyanidins (non-hydrolyzable tannins) are oligomers or polymers of flavan-3-ols that can undergo thermal degradation, depolymerization, and polymerization. Due to the high temperatures (>60 °C) in drying, they can result not only in their condensation to insoluble complex tannins but also in their decomposition into monomers by acidic hydrolys. In short, severe drying conditions (high temperature and long process times) may destroy flavanols, whereas mild drying or combined methods could preserve or depolymerize the polymeric procyanidins into oligomeric procyanidins or monomers [97].

#### 3.5.3. Effects of Drying Technology on the Concentration of PCs

Research has been developed to evaluate the PC retention after drying. De Brito et al. [102] found that the content of phenolic compounds decreased by 32% when solar drying was applied. In the same way, Cruz et al. [103] found a severe reduction in phenolic compounds after solar drying that varies according to the cultivar evaluated. The reduction of epicatechin was between 55 and 68%, while for catechin the reduction was between 35 and 50%. Additionally, Alean et al. [36] reported a reduction of 45% in phenolic compounds after applying drying with hot air at 40 °C. On the other hand, it was observed that a high content of phenolic compounds was reached by applying the minimum level of processing to cocoa. Dumarche et al. [104] proposed a methodology to preserve the phenolic compounds in cocoa beans with the aim of enhancing the functional potential and producing attractive colors (red and purple) of cocoa derivatives for consumers. The method stated by Dumarche et al. [104] established that the unfermented cocoa seeds should be processed under controlled conditions using acidic agents for the acidification (i.e., citric, tartaric, lactic, acetic, phosphoric acids, etc.), a pH of from 1 to 3, for a period of about 3 to 6 h at a temperature of 5 to 30 °C, and then the seeds were dried at a temperature between 60 and 80 °C with an infra-red heater. The authors stated that by applying this methodology, cocoa nibs had a high content of phenolic compounds of at least 20 up to 60 mg equivalent of epicatechin/g of dried cocoa. These findings are in line with the results of Faborode et al. [82], who observed that when drying temperatures are used below 60 °C, the retention of phenolic compounds is higher, and therefore the functional potential is preserved.

Furthermore, according to the results obtained by Di Mattia et al. [96], a sharper decrease in the procyanidins monomers (P1) based on (-)-epicatechin and (+)-catechin was observed, which was the fraction most susceptible to degradation during solar drying. These findings are attributed to the fact that, during the first stage of solar drying, the condensation reactions are developed, but then the faster moisture loss decreases the enzymatic activity, and the diffusion and migration are difficult, limiting any further procyanidin degradation. In general, procyanidins decreased significantly during drying with a reduction of 65 to 70%. P1 monomers significantly decreased (*p* < 0.01) up to 30%. Procyanidin pentamers (P5) decreased from 9% to 7%. On the other hand, the procyanidin dimers-tetramers (P2–P4) did not show any significant changes, while the P7–P10 fractions increased in the last steps of the drying process due to oxidation and polymerization reactions. Due to the severe reduction of phenolic compounds caused by drying, more research should focus on studying the effect of drying technology on the profile of procyanidins on their reactions and functional activity, given the multiple biochemical reactions that can occur and their interaction with other components such as proteins and carbohydrates.

Exploring the interactions and potential synergistic effects among different phenolic compounds is essential for a comprehensive understanding of their influence on sensory attributes in cocoa. Recent research has indicated that specific phenolic compounds can interact with other cocoa compounds in ways that increase sensory perceptions, such as astringency, bitterness, and color. Ramos-Pineda et al. [105] demonstrated that cooperative interactions between catechin and epicatechin in binding to proteins, when both types of flavanols are present, may elucidate the synergies observed in astringency. Synergistic effects are also evident among various phenolic compounds, which may contribute to the diverse qualitative manifestations of astringency. Furthermore, this phenomenon may clarify why the qualitative composition of phenolics has a more significant impact on astringency than the overall concentration of these compounds. Similarly, Ferrer-Gallego et al. [106] demonstrated that there is a synergistic astringent effect between phenolic compounds named the ‘co-astringency’ effect, consisting of the increase observed in the astringency when they are tasted together versus individually at the same final concentrations. The mixture of catechin and epicatechin showed an increase in the perception compared to individual compounds, and a synergistic effect between volatile compounds and phenolic compounds resulted in the increase in the rate intensities of the astringent perception. Additionally, studies have shown that the presence of specific phenolic compounds, such as epigallocatechin, is used to establish a color-composition correlation. The color development in cocoa products, which varies from the typical brown color to a red-pink color (it produces ruby chocolate), influences consumer acceptance and market value [107].

In recent years, research has focused on the effects of cacao drying methods on the retention and profile of phenolic compounds, emphasizing their importance for flavor and nutritional quality. Previous works revealed that hot air drying resulted in a significant reduction of phenolic content compared to solar drying. Additionally, Santhanam Menon et al. [63] analyzed the impact of freeze drying on cacao beans and reported that this method retained a higher quantity of phenolic compounds compared to oven drying, adsorption drying, and vacuum drying, thus preserving the functional potential.

As noted, the optimization of the conditions of the drying process in the research studies aims to produce desirable traits regarding the functional potential and the flavor profile of the final product considering the target market. Therefore, there are markets, such as functional food and beverages, pharmaceuticals, and cosmetics, that demand cocoa beans with high flavonoid concentration due to the importance of key health and wellness insights in product development. On the other hand, there are confectionary markets that may require cocoa beans with acceptable concentrations of phenolic compounds that produce moderate levels of astringency and bitterness to produce gourmet chocolate bars with balanced sensory profiles.

### 3.6. Short-Chained Carboxylic Acids: Crusting Phenomena, Acidity Perception, and Physicochemical Changes During Drying

Acetic acid is produced primarily by the oxidation of ethanol in the presence of oxygen by acetic acid bacteria during fermentation. It migrates and is stored inside the seeds. Later, during drying, this acid is evaporated along with the moisture due to its volatile nature [108]. On the contrary, the lactic acid produced by the lactic acid bacteria during fermentation is stored inside the seeds, and it cannot be easily removed as it is not volatile. However, Camu et al. [109] found that the release of lactic acid is possible if the drying process is slow, since it can be partially transported by water from inside the bean to the surface. The desirable range for acetic and lactic acid concentrations during fermentation is typically between 0.5 to 1.5% (*v*/*v*) and 0.1 to 0.8% (*v*/*v*), respectively. Higher concentrations, above 2% of acetic acid and above 1% for lactic acid, can produce excessive sourness, negatively impacting the overall quality of cacao [110].

The mass and heat transfer are controlled by the conditions of the boundary layer of the surface of the cocoa beans to be dried. The difference between the testa and the cotyledon cellular structures would be responsible for the difference in moisture diffusivity inside the cocoa seed. When high temperatures or high rates are used during drying, the continuous diffusion or capillarity of water and acids inside the seeds can be interrupted. In these conditions, the testa will generally dry faster than the cotyledon, and as the testa adheres to the cotyledon, it produces the crusting phenomena. In consequence, the premature drying of the testa, or the testa hardening, contributes to slowing down the moisture and acid outward migration from the seeds, leading to the retention of acids, and therefore negatively affecting the flavor. The intense acidic taste produced cannot be removed even in the subsequent cocoa processing stages. Furthermore, the crusting phenomena reduce the availability of oxygen inside the seeds, limiting the enzymatic oxidation reactions [111].

To effectively control the crusting phenomenon and enhance the evaporation of lactic acid during cocoa drying, several strategies and emerging technologies can be employed. One promising approach involves utilizing intermittent drying techniques, which incorporate cycles of heating and resting, allowing for better moisture management and reducing the risk of crust formation that impedes lactic acid evaporation [12]. Additionally, the application of microwave-assisted drying has shown potential in promoting uniform heating, thereby minimizing crust development while simultaneously facilitating the release of volatile compounds [112]. Vacuum drying is another effective method that operates under reduced pressure, which enhances the evaporation of lactic acid without the need for excessive heat, helping to preserve the sensory qualities of cocoa [35]. Furthermore, specially designed drying chambers with optimized airflow management can significantly improve the evaporation rates of short-chained carboxylic acids during the drying process. These drying technologies represent an integrated approach to improving drying efficiency while maintaining the biochemical integrity of cocoa.

As acidity plays a crucial role in the flavor quality, and elevated concentrations may be detrimental for product acceptability, producing low sensory quality in dried cocoa beans, research was carried out to study the effects of cocoa drying on acids. To do this, Jinap et al. [113] studied the effects of oven-drying at 60 °C, air-blow drying in a flatbed dryer, shade drying (drying in the shade for 120 h and then drying in a ventilated oven at 60 °C for 10 h), and solar drying on the titratable acidity, volatile fatty acids (VFA), such as acetic, propionic, butyric, isobutyric, and isovaleric acids, and the sensory attributes of chocolate derived from dried beans. A steadily decreasing trend over time was observed after the application of all drying methods. Beans dried at 60 °C in a ventilated oven exhibited high acidity and the highest total VFA content (3206.02 mg/kg), while the shade-dried samples had less acidity due to putrefactive microbial activity induced by the prolonged high bean moisture. On the contrary, solar and air-blown drying methods generated lower levels of VFA (751.00 mg/kg and 811.55 mg/kg, respectively). Additionally, the expert sensory panel considered that chocolate from air-blown dried beans had no off-flavors, and sun-dried chocolates were characterized by a fruity and well-developed chocolate flavor. In the shade-dried chocolate, undesirable attributes were perceived, such as musty, rancid, cheese rind, hammy, and unpleasant.

Furthermore, research has focused on studying acid mass transfer properties during drying, considering the differences between the testa and the cotyledon. Therefore, Nganhou et al. [83] studied the migration of water and acetic acid through a microanalysis method to determine the profiles of water and acetic acid concentrations in different layers of the bean from the center to the surface/testa. An increase in the coefficient of transportation during drying time was observed. Furthermore, the diffusion coefficients of acetic acid and water are higher at the surface/testa than in the center of the cocoa bean, and the authors stated that these findings contribute to explaining the occurrence of crusting phenomena.

The acid mass transfer properties during drying continued to be studied in order to control the process conditions that produce the desired contents of moisture and volatile acidity in the dried cocoa beans with minimal energy resources. To do this, García-Alamilla et al. [38] evaluated a theoretical model that considered water, acids, and heat transfer during cocoa drying. It was found that the acidity transport rate decreases over drying time, and a slow rate of acidity transport was observed. The testa moisture reaches a value of 0.1 in 1 h of drying, and therefore it acts as a barrier for acidity transport. The model predicted that with low diffusivity, the acidity can evaporate from the interface of cocoa testa to the air, but only 40% of the initial acidity can be removed after 22 h at 80 °C. In terms of process efficiency, the authors stated that drying at 70 °C is optimal as cocoa is processed at a lower temperature to avoid quality losses that allow the acidity to be removed at a similar rate to 80 °C but also reduces energy consumption by 77%.

After that, considering that high concentrations of VFA are detrimental to the quality of cocoa derivatives, Páramo et al. [114] studied the drying kinetics for moisture content and VFAs associated with undesirable vinegar and rancid sensory attributes, such as acetic, propionic, butyric, isobutyric, and isovaleric acid concentrations in cocoa beans during drying. Temperatures at 40 and 60 °C, air velocities of 0.5 and 2 m/s, and cocoa beans with and without testa were considered. It was found that, after drying, the beans with testa had higher concentrations of water and VFA than the beans without testa. Furthermore, higher diffusivities were obtained for beans without testa at both temperatures, which demonstrated that the testa produces a significant mass transfer resistance for water and VFA. The reduction of diffusivities with testa was in the order of 1/3 to 1/9 with respect to diffusivities without testa. Propionic and butyric acid diffusivities were statistically lower than water diffusivity (1/6 and 1/22, respectively) at both temperatures and in beans with or without testa. Consequently, acidity losses during drying were 10 times slower than moisture lost with or without testa. To reduce acidity in dried beans, the authors proposed intermittent drying with steps without airflow that allow the rearrangement of acidity and water profiles inside the beans.

### 3.7. Methylxanthines: Its Association with Bitterness and Changes During Drying

Cocoa beans contain approximately 3 to 4% methylxanthines or purine alkaloids. Theobromine (3,7-dimethylxanthine) is the main xanthine, with 2 to 3%, followed by caffeine (1,3,7-trimethylxanthine) in low quantities that can vary between 0.2 and 0.6%, and theophylline found as traces [15]. These alkaloids are closely associated with the bitter perception of cocoa derivatives [15,113]. In addition, Luna et al. [94] reported positive statistical correlations between theobromine and bitterness and a green flavor note.

Despite methylxanthines not undergoing complex biochemical transformations during drying, it was found that these chemical compounds are significantly reduced during drying, depending on the genotype and variable conditions. Theobromine and caffeine are lost after drying through diffusion and migration from cotyledons [115]. In general, the suitable concentration of caffeine typically ranges from 0.5 to 2.5%, while theobromine should fall within the range of 1 to 3% [116]. This range contributes to generating balanced flavor profiles in which bitterness, a core attribute in cocoa, can be evaluated at a mild level, thereby allowing one to perceive the fine aromas.

The variations in methylxanthine concentrations during the drying of cocoa can be attributed to several proposed mechanisms that reflect the impact of different drying methods on these bioactive compounds. High drying temperatures associated with conventional methods, such as hot air drying, may induce thermal degradation of methylxanthines, leading to significant reductions in their concentrations due to the denaturation of the associated cellular structures and the breakdown of these compounds [77]. Conversely, gentler drying techniques, such as freeze drying and vacuum drying, preserve the structural integrity of cocoa beans and the stability of methylxanthines by avoiding excessive thermal stress, thus resulting in higher retention levels of these compounds [35]. Furthermore, differences in moisture content during the drying process can influence the degradation kinetics of methylxanthines, as a high moisture environment may promote the hydrolysis of these compounds, leading to decreased concentrations [117].

Furthermore, the influence of the drying process and equipment on methylxanthine concentration has been studied. Peláez et al. [115] reported that theobromine and caffeine concentrations significantly decreased after solar drying from 1.449 to 1.140 and from 0.410 to 0.165 g/100 g db, respectively, which implies a reduction of 20% and 60%, respectively. In contrast, Cruz et al. [103] observed that methylxanthines did not suffer changes after solar drying, and there were no significant differences between caffeine or theobromine concentrations after drying of fermented cocoa seeds.

In addition, Deus et al. [77] evaluated the application of different drying equipment on the concentration of methylxanthines. The authors used: i. a dryer with a stainless steel platform and plastic roof with UV protection; ii. an artificial dryer using a wooden platform with an artificial heat source; iii. a traditional dryer in a barge with a wooden platform and direct sunlight; iv. a mixed dryer with a stainless steel platform and mobile plastic roof with UV protection for drying coverage and exposure to the sun. Fermented cocoa seeds prior to drying exhibited higher theobromine (19.44 mg/g) and caffeine (2.74 mg/g) concentrations than dried cocoa beans, regardless of the dryer used. Significant reductions of theobromine and caffeine after drying between 15 and 42% and 23 and 48%, respectively, were observed. The traditional drying method retained mostly methylxanthines and produced the minimum losses in theobromine (15%) and caffeine (23%), respectively.

The study highlighted contradictory findings regarding the impact of drying on the methylxanthine content, with some studies reporting significant reductions while others observed no changes. The studies collectively indicate that while drying conditions (temperature, duration, and method) significantly influence methylxanthine stability, these effects are mediated by the initial composition of the beans and the specific drying parameters applied. Recognizing these shared findings provides a foundation for reconciling discrepancies in the reported results. Future research should aim at standardizing drying conditions and analytical methods to ensure comparability and provide definitive insights into the effect of drying on methylxanthine content.

### 3.8. Volatile Aromatic Compounds: Amadori Heyns Compounds and the Onset of Maillard Reactions

The cocoa flavor is defined by volatile and non-volatile compounds that are perceived by the combined impression of taste and aroma [118]. The flavor of cocoa develops throughout the different stages of post-harvest processing (Figure 6). The genotype conditions the flavor formation in accordance with the initial composition of the fresh cocoa seeds: components such as proteins and carbohydrates produce the flavor precursors, phenolic compounds, and methylxanthines, as mentioned before, are directly responsible for the astringency and bitterness sensations perceived in cocoa derivatives.

Moreover, in addition to the genotype, flavor is also influenced by the following cocoa processing steps: i. Fermentation, in which the flavor precursor metabolites, such as amino acids, peptides, and reducing sugars, are produced from the seed’s components by the action of the endogenous enzymes (proteases, invertases), which act according to the conditions of temperature and pH. On the other hand, the activity of the bioprocess mediating microorganisms (especially yeasts) generates volatiles with desirable aromatic notes (i.e., fruity, floral). ii. Drying plays a key role due to key intermediate volatile molecules: Amadori Heyns compounds and volatile compounds are generated from the interactions of the flavor precursor metabolites. Through the drying process, a significant decrease in free amino acids, peptide-N, and total reducing sugars was observed, and in parallel, an increase in the concentration of trimethyl-, tetramethylpyrazine, and total pyrazines in cocoa beans was found [119,120]. Hashim et al. [120] reported that when the drying temperature increased from 40 to 80 °C, the concentration of free amino acids decreased sharply, which may indicate that these are consumed in the mild/premature Maillard non-enzymatic browning reactions. Drying at 40 °C resulted in high concentrations of reducing sugars (1.72 g/kg) and acidic, hydrophobic, total, and other free amino acids at 1.05, 5.35, 11.41, and 5.01 g/kg, respectively. The authors explained that this temperature (40 °C) facilitates the invertase activity and proteolysis by aspartic proteinase and carboxypeptidase enzymes to generate flavor precursors, since these enzymes have an optimum activity at around 45 °C.

At the same time, drying reduces but does not totally eliminate volatile organic acids that, in excess, are detrimental to the flavor because they are related to undesirable aromas, such as vinegar-like, rancid, and pungent notes produced by propanoic, propionic, butyric, and isobutryic acids [114]. Furthermore, the thermal flavor related to the volatile compounds generated in drying and roasting is due to the non-enzymatic reactions of Maillard, which involve the flavor precursor metabolites formed during fermentation. The carbonyl groups of reducing sugars and the amino groups of peptides and free amino acids interact to produce a diversity of volatile compounds and colored polymer products [121,122]. Maillard reactions depend on the temperature and the water activity. It is complex to control the rate of the reaction and the reaction products. In practice, the Maillard reaction occurs most rapidly at temperatures higher than 100 °C (achieved mainly during roasting) and at intermediate water activity values (0.5–0.8) [123]. Although, as has been reported, there is an increase in volatile compounds in cocoa beans as the drying temperature is increased up to 80 °C [119]. Owusu [124] and Hashim et al. [120] reported that volatiles are generated even at low temperatures of between 30 to 50 °C, and the Maillard reactions are initiated and promoted by a drop in moisture content during drying.

When cocoa is subjected to heat treatment during drying, Maillard reactions occur, and major stable Amadori Heyns compounds are produced that contribute to flavor formation in cocoa. Condensation is also observed, as well as amino acids with either glucose or fructose producing rearrangement. Throughout the drying time, the level of Amadori compounds is raised, and the roasted aroma intensifies [125]. Furthermore, these Amadori compounds are key intermediates in later stages of cocoa bean-to-bar processing, such as roasting, because in this stage they generate multiple volatile compounds [126]. For this reason, the Amadori compound product that results from the interaction between fructose and phenylalanine appears to be an intermediate or a precursor of particular interest, as it can participate in reactions that generate numerous phenyl derivatives responsible for cocoa-like flavor after high temperatures during roasting.

Approximately 400–600 volatile compounds have been identified in cocoa beans. The principal chemical groups of compounds are pyrazines, esters, ketones, pyrroles, furans, furanones, pyrones, terpenes, and alcohols [127]. Pyrazines are the key type of heterocyclic volatiles and the main components forming the cocoa aroma. They exhibit cocoa, nutty, and roasty aromas, so these molecules are highly desirable in cocoa beans [127]. Esters, ketones, aldehydes, and some alcohols are favorable for cocoa flavor quality because they are associated with fruity and flowery aromas [128]. While furans, furanones, and pyrroles are responsible for desirable aromatic notes, such as chocolate, sweet, and caramel [129]. On the contrary, high concentrations of phenols negatively affect the aroma, since they are related to undesirable off-flavors, such as smoky and hammy, which are characteristic of insufficiently dried beans or rain-soaked beans [130]. All volatiles present in cocoa have an associated odor threshold value (OTV) that allows for the calculation of the Odor Activity Value (OAV), which is a measure used to evaluate the potential contribution of volatile compounds to the overall aroma and flavor profile of cocoa. The OAV is considered significant if it is greater than 1, indicating that the compound’s concentration in the product is sufficient to influence aroma perception. We mention the specific odor threshold value in the key volatiles that showed significant changes after drying processing [131].

When sensory analysis was carried out to assess the effect of drying methods, such as solar drying, air-blowing, shade drying, and oven drying, on the flavor of cocoa, it was found that the solar-dried cocoa beans rated higher in cocoa flavor and exhibited fewer off-flavors [92,113]. During the drying process, flavor precursor metabolites are converted into two main classes of active flavor components, pyrazines and aldehydes. It was observed that fermented solar-dried cocoa beans were characterized by having a high diversity of compounds, such as esters, aldehydes, pyrazines, alcohols, acids, and furans [132]. It has been demonstrated that when flavor precursor metabolites are present in lower concentrations due to the fact that fermentation was not applied, dried unfermented cocoa beans presented a lower number of volatiles, and these were mainly alcohols, ketones, hydrocarbons, and other compounds, such as nitrile, pyridine, ester, and lactone. On the other hand, during drying, moderate temperatures and relatively high moistures produce the formation of furanones and pyrans by the degradation of monosaccharides [132].

When the dynamics of volatile compounds were analyzed through solar drying, it was found that the concentration of alcohols (2,3-butanediol with OTV = 668 mg/kg and 1,3-butanediol associated with sweet, creamy, slightly buttery, and fruity aromas), esters (3-methyl-1-butanol acetate related to fruity, banana-like aroma, ethyl acetate with OTV = 0.94 µg/g and fruity aroma, and isobutyl acetate, fruity aroma), and pyrazines (tetramethylpyrazine with OTV = 38 µg/g, earthy) increased during the solar drying process. However, the concentration of acids (isobutyric with OAV = 0.19 mg/kg, isovaleric with OTV = 0.041 mg/kg, hexanoic with octanoic and nonanoic acid associated with rancid, cheese, dairy, acidic, sour, pungent, stinky, and fatty flavors), aldehydes (pentanal and phenylacetaldehyde), and ketones (2,3-butanedione: creamy, buttery, pungent, caramellike aroma with OTV = µg/g, acetoin: buttery aroma and OTV = µg/g) decreased [39,133].

Utrilla-Vázquez et al. [132] reported that solar drying generates key aromas and technological markers of the process. These are desirable compounds, such as 2-phenylethyl acetate (OTV = 0.137–0.233 µg/g, honey-like aroma), 2-heptanol (OTV = 0.01 µg/g, citrus, fresh, lemon aroma), phenylethyl alcohol (floral, spicy, and honey aroma), and acetoin (buttery aroma) that were at higher concentrations, while the 3-methylbutanal (OTV = 0.0054–0.08 µg/g, malty and chocolate aroma) and benzeneacetaldehyde (OTV = 0.022–0.154 µg/g, honey-like aroma) were found at low concentrations. Furthermore, pyrazines are key compounds generated due to Maillard reactions, which are initiated by a reduction in moisture content and temperatures between 30 and 50 °C. Pyrazines presented variations in concentration. In this sense, the quantity of trimethylpyrazine (OTV = 0.29 µg/g, cocoa, roasted nuts aromas) and tetramethylpyrazine (OTV = 38 µg/g, earthy) was higher than methylpyrazine (OTV = 27 µg/g, buttery), 2–3 dimethylpyrazine (OTV = 0.123 µg/g, caramel, cocoa aromas), 2–5 dimethylpyrazine (OTV = 2.6–17, chocolate, nutty aromas), and 2–6 dimethylpyrazine (OTV = 1.021–8 µg/g, nutty, herbal aromas).

Other authors reported that when the drying temperature increases, the degradation of some desirable volatile compounds, such as pyrazines, increases due to the fact that the convective phenomena are stronger [36]. Additionally, Rodriguez-Campos et al. [133] observed that after applying convective cocoa drying in an oven at 70 and 80 °C after 6 days of fermentation, it resulted in a volatile profile similar to that obtained by solar drying.

The reported information regarding the temperature at which the Maillard reaction significantly influences the formation of volatile compounds during drying lacks uniformity. This discrepancy may stem from differences in experimental setups, variations in the composition of cocoa beans, or the methodologies employed to analyze volatile compounds. To address these discrepancies, it is necessary to identify commonalities in the reported temperature ranges, focusing on the interaction of drying temperature, time, and flavor precursors to metabolite concentration.

In addition to temperature and moisture content, several other factors significantly influence the formation and degradation of volatile compounds during the drying of cocoa. The choice of drying method plays a critical role, as techniques such as convective drying can enhance heat transfer while potentially leading to the loss of delicate volatile compounds through excessive evaporation [134]. Airflow rate is another essential factor; increased airflow can promote rapid moisture removal but may also facilitate the loss of volatile aromas [26]. The duration of drying is equally important, as prolonged exposure to heat can result in the degradation of sensitive volatiles, while shorter drying times may not effectively remove moisture, potentially leading to off-flavors [2]. Additionally, the initial biochemical composition of the cocoa beans, including their variety and polyphenolic content, can significantly affect the profile of volatile compounds produced during the drying process. The presence of specific enzymes, such as lipoxygenase and alcohol dehydrogenase, can also influence the breakdown and synthesis of volatile compounds during drying [135].

#### Physical Phenomena: Diffusion and In Situ Oxidative Processes

The phenolic compounds during drying can be affected especially by the air temper-ature, airflow, and process time. The drying temperature plays a key role in the retention of phenolic compounds. When the temperature increases, the content of phenolic com-pounds decreases due to their thermolabile nature. Furthermore, at high temperatures and over long drying times, degradation can also be affected by cell destruction [36,96].

Bean moisture is another key variable due to the interaction between phenolic com-pounds and water. The polarity of phenolic compounds allows them to dissolve and mi-grate to the surface [36]. During drying, the mass transfer of PCs occurs through molecular diffusion under the influence of vapor partial pressure that diffuses from the cocoa surface into the moving air drying. The PC diffusion from the surface of cocoa beans occurs with the evaporating water, which is controlled by the drying rate [96]. Contrary to this, Alean et al. [71] studied the polyphenol oxidation process in situ during the drying of cocoa beans by using microwaves. They reported that the degradation of polyphenol compounds during drying does not obey a transport mechanism, such as diffusion, as it had always been stated. In contrast, it is due to in situ oxidative processes. The authors proved that the in situ oxidation of polyphenols during this process is generated as the water evaporates, followed by the penetration of air oxygen to the bean that oxidizes the polyphenols.

## 4. Conclusions

Cocoa drying is a critical post-harvest operation that complements fermentation, with both processes playing a decisive role in determining the final quality of cocoa and its derivatives. This process involves complex phenomena of heat and mass transfer, along with biochemical transformations and chemical reactions. These interactions directly affect parameters such as moisture content, the stability of bioactive compounds, and the sensory attributes of cocoa. Key compounds influenced during drying include short-chain carboxylic acids, which contribute to acidity (the optimal levels to ensure balanced flavor profiles are less than 2% and 1% for acetic and lactic acids, respectively), flavonoids, associated with astringency and bitterness (with optimal ranges for total phenolic compounds between 700 to 1300 mg GAE/100 g db, and for flavanols is 300 to 700 mg of epicatechin); methylxanthines related to bitterness (optimal value for caffeine is 0.2% db and for theobromine is 3% db); and volatile compounds, essential for cocoa’s characteristic aromas. A deep understanding of these phenomena is fundamental to ensuring a drying process that not only preserves but also enhances the sensory and functional properties of the final product.

Challenges related to cocoa drying technology include the design and evaluation of equipment that maximizes thermal efficiency, minimizes the loss of bioactive compounds, and allows precise control of key operational variables, such as temperature, relative humidity, airflow velocity, and drying time. In this regard, the use of mathematical models and simulations based on computational fluid dynamics (CFD) stands out as a powerful tool for optimizing both the drying kinetics and the uniform energy distribution within dryers. However, it was observed that for a comprehensive analysis, it is essential to integrate these techniques with sensory evaluations of cocoa derivatives, such as chocolate, butter, and powder, to correlate drying conditions with organoleptic and functional attributes.

Research into the effect of drying on phenolic compounds, organic acids, and volatile aromatic compounds and their sensory implications underscores the importance of selecting appropriate drying methods tailored to specific cocoa materials and final product requirements. The review highlights that traditional drying methods may compromise the retention of desirable volatile compounds, thereby negatively impacting the overall flavor profile. Emphasizing alternatives such as intermittent drying has emerged as a promising strategy, as it allows for the optimization of drying parameters—such as temperature, airflow, and moisture content—thus enhancing the preservation of key volatile and phenolic compounds. Furthermore, innovative methods such as solar and hybrid drying systems provide effective solutions to mitigate crusting and ensure uniform moisture distribution, ultimately leading to the development of high-quality cocoa products while addressing market demands for enhanced flavor and aroma characteristics.

The examination of various drying technologies reveals critical insights into their cost-effectiveness, scalability, and resource availability, all of which are essential considerations for enhancing the practical application of research findings in the cocoa sector. Solar drying methods, such as sliding surface dryers and greenhouse-like systems, represent cost-efficient options for small-scale farmers, utilizing readily available materials and renewable energy. These methods not only reduce operational costs but also minimize environmental impact, making them an attractive choice for rural communities with limited financial resources. However, challenges such as inadequate protection against adverse weather conditions must be addressed to improve reliability and consistency in drying outcomes.

In contrast, advanced drying systems—including hybrid solar dryers, fluidized bed dryers, and heat pump technology—offer enhanced efficiency and better control over drying parameters. While initially more expensive and complex to implement, these systems can provide significant long-term benefits by improving product quality and reducing drying times. The integration of advanced technologies can facilitate the scaling of operations, providing cocoa producers with the ability to meet market demands for high-quality products while ensuring the retention of bioactive compounds.

Furthermore, the application of intermittent drying presents a promising alternative to conventional methods. By optimizing drying conditions through controlled heating and resting periods, this technique not only enhances the chemical and sensory quality of cocoa but also offers opportunities to reduce energy consumption. The combination of traditional knowledge with innovative drying technologies can lead to a more sustainable and economically viable alternative. Ultimately, the choice of drying method and equipment must consider local conditions, resource availability, and market demands to ensure that cocoa producers can implement effective practices that improve quality while remaining economically viable.

Given the growing demand from the food, confectionery, pharmaceutical, and cosmetic markets for high-quality products, post-harvest operations such as drying must be carefully managed to achieve specific sensory and functional profiles that meet these requirements. This study highlights the importance of adopting advanced technologies in dryer design, considering both energy efficiency and sustainability, to ensure that the resulting cocoa beans exhibit balanced flavor characteristics and a high concentration of bioactive compounds, such as polyphenols and methylxanthines, which positively impact human health and the product’s commercial value.

## 5. Recommendations for Future Research

Despite advancements in drying technology, significant knowledge gaps remain, limiting a comprehensive understanding and optimization of this process. Consequently, we have delineated potential actions to address the identified research gaps across various areas of focus.

The first area involves advanced modeling techniques, particularly the integration of multiphysics models, which includes the utilization of Finite Element Methods (FEM) for structural analysis and Computational Fluid Dynamics (CFD) for modeling heat and mass transfer during cacao drying. FEM allows for the analysis of heat distribution and mechanical stresses in cacao beans, essential for preserving product quality, while CFD facilitates the understanding of airflow patterns and moisture movement, thereby optimizing drying parameters for enhanced energy efficiency and quality retention. The development and validation of these models require various data inputs, including the physical properties of cacao (thermal conductivity, specific heat capacity, and moisture diffusion coefficients), geometrical configurations of the drying setup, boundary conditions (inlet air temperature and humidity), and experimental validation data from pilot-scale drying tests [136]. We recommend establishing collaborations with research institutions specializing in food engineering for model development and validation, which will provide critical data inputs for optimizing drying conditions.

We suggest exploring the impact of different airflow profiles and thermal heterogeneity within drying systems on the sensory quality of cocoa. This research could involve partnerships with local universities and industry stakeholders to conduct field trials and sensory evaluations, directly linking drying conditions with product quality.

Further investigation into optimizing the retention of bioactive compounds during drying, especially for cacao aimed at functional or nutraceutical markets, should be prioritized. Collaborations with food technology research centers can facilitate this research, focusing on methodologies that correlate drying parameters with sensory and functional performance.

Lastly, given the increasing urgency to address environmental challenges, we recommend studying the feasibility of renewable energy sources in drying systems, such as hybrid solar models. This includes partnerships with agricultural extension services to promote sustainable practices that reduce the carbon footprint of cocoa processing while improving sensory quality and the preservation of the functional potential.

## Figures and Tables

**Figure 1 foods-14-00721-f001:**
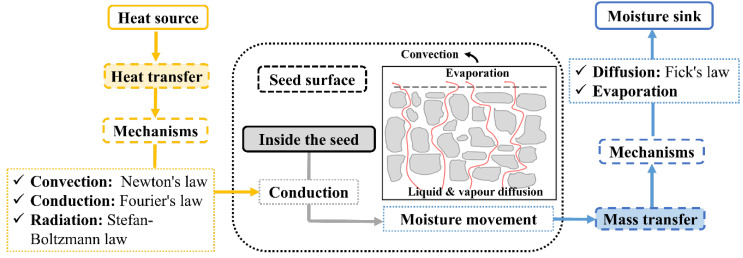
Schema of mass and heat transfer phenomena and associated mechanisms and laws in cocoa drying.

**Figure 2 foods-14-00721-f002:**
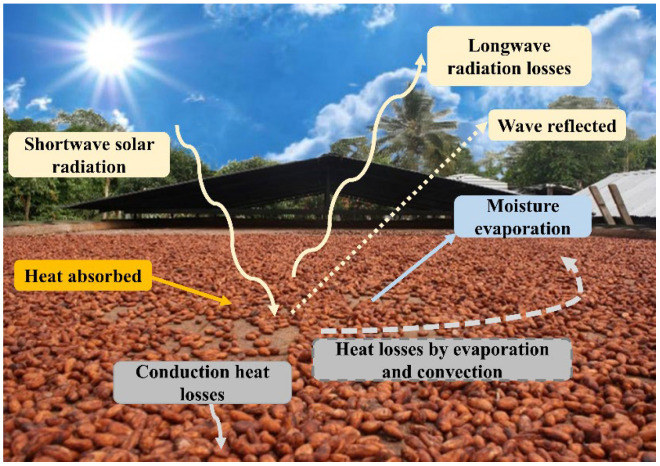
Solar drying: operating principle and mass and heat transfer mechanisms. Source: Adapted from Sharma et al. [42].

**Figure 3 foods-14-00721-f003:**
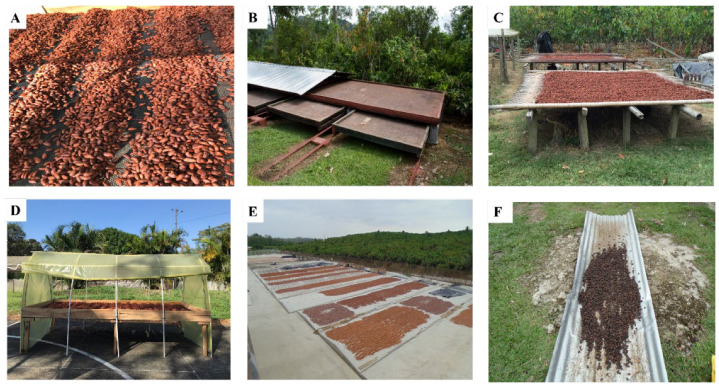
(**A**) Method of arrangement of cocoa seeds during the first hours of drying. Solar drying equipment commonly used by cocoa farmers: (**B**) Sliding surface dryers (elbas), (**C**) drying surface made of dry canes (*Gynerium sagittatum*), (**D**) greenhouse-like drier, (**E**) cement drying surface, (**F**) zinc drying surface.

**Figure 4 foods-14-00721-f004:**
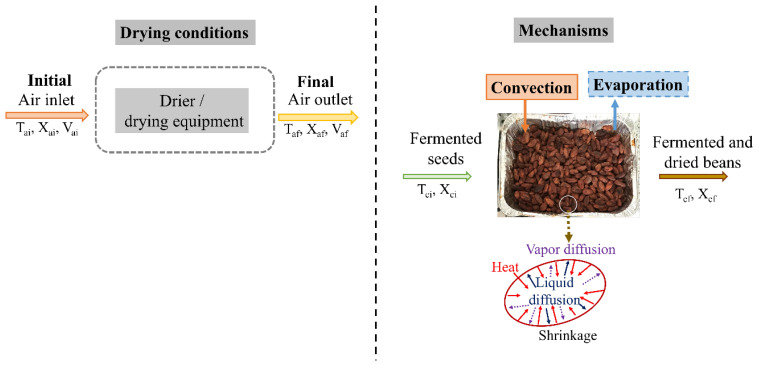
Convective drying: principle and mass and heat transfer mechanisms. Subscripts ai and af refer to initial and final air variables, respectively. Subscripts ci and cf refer to initial cocoa seed and final cocoa bean variables, respectively.

**Figure 5 foods-14-00721-f005:**
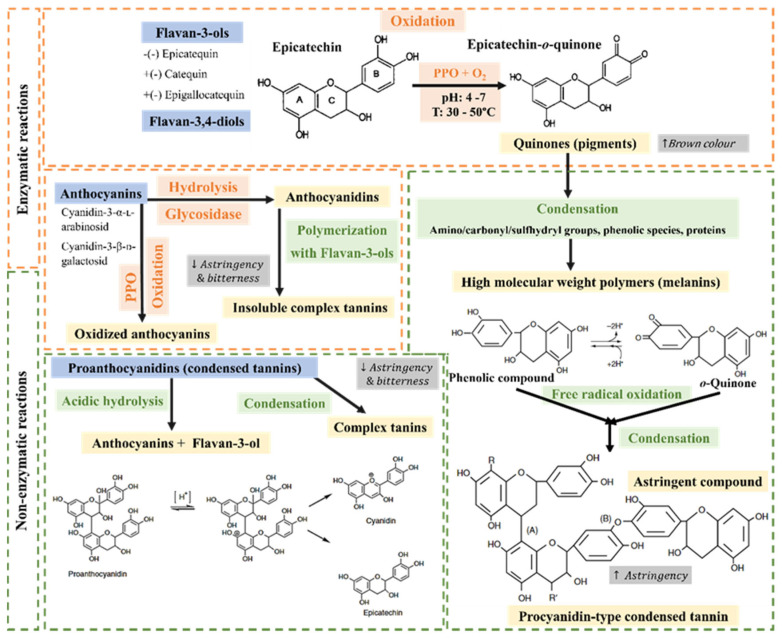
Schema of enzymatic and non-enzymatic reactions of phenolic compounds during drying. The orange color represents enzymatic reactions, while the green color represents non-enzymatic reactions. The flavonoids in cocoa are highlighted in blue and the products of the reactions in yellow. Source: the examples of chemical reactions were taken from Damoradan et al. [8].

**Figure 6 foods-14-00721-f006:**
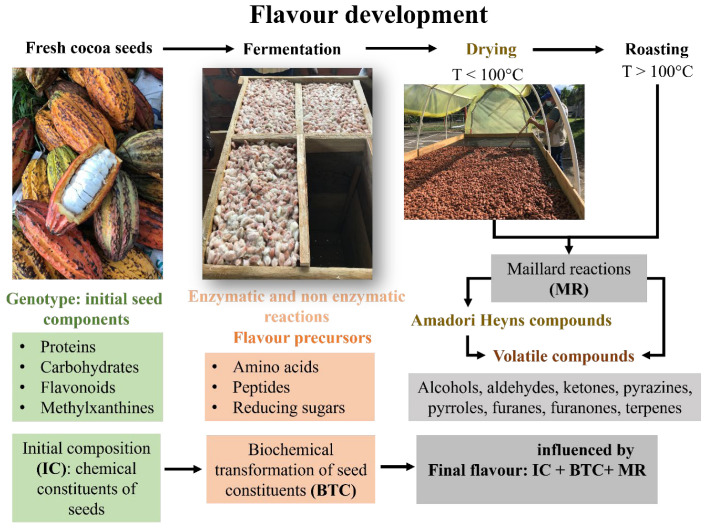
Schematic diagram illustrating cocoa flavor development, considering the influences of genotype and fermentation, as well as the roles of drying and early Maillard reactions, which produce volatile compounds responsible for flavor sensory attributes.

**Table 1 foods-14-00721-t001:** Overview of Solar Drying Methods.

Type of Solar Dryer	Design	Description	Advantages	Disadvantages
(A) Cocoa Seed Arrangement Method During the First Hours of Drying	Uniform distribution of seeds in thin layers to optimize sun exposure.	Seeds are evenly distributed to optimize sun exposure and improve drying uniformity.	Better control over heat distribution, prevents clumping.	Requires constant monitoring to prevent uneven drying.
(B) Sliding Surface Dryer (Elbas)	Inclined surface allowing gradual movement of cocoa during drying.	Inclined surface allowing controlled movement of cocoa to enhance drying consistency.	Enhances drying uniformity, reduces the need for frequent turning.	Higher initial investment and maintenance required.
(C) Drying Surface Made of Dry Canes (Gynerium sagittatum)	Base made of interwoven dry canes to facilitate airflow and prevent moisture accumulation.	Structure made of dry canes that improves ventilation and reduces moisture buildup.	Good airflow circulation, prevents moisture buildup and mold formation.	Lower durability, requires frequent material replacement.
(D) Greenhouse-like Dryer	Closed structure with a transparent cover to retain heat and protect against rain.	Closed chamber with a transparent cover maintaining stable temperatures and preventing rain interference.	Controlled temperature, less dependence on external weather conditions.	Higher installation cost and periodic maintenance required.
(E) Cement Drying Surface	Elevated cement platform providing stability and ease of handling cocoa.	Cement surface where cocoa is spread in thin layers for efficient drying.	Easy to clean, provides a stable drying base.	Can retain moisture if not properly managed.
(F) Zinc Drying Surface	Metal sheet surface that absorbs and transfers heat quickly to accelerate drying.	Zinc surface that retains and transfers heat rapidly, accelerating the drying process.	Rapid heating, accelerates moisture removal.	Risk of overheating and uneven drying.

**Table 2 foods-14-00721-t002:** Main DUCC drying methods.

Type of Dryer	Design and Mechanism	Performance	Energy Efficiency	Impact on Cocoa Quality	Main Advantages	Main Disadvantages
Hot-Air Oven Dryer	Forced convection of hot air generated by electric resistances or gas.	Drying time of 24–48 h, variable capacity.	High efficiency at moderate temperatures.	May lead to loss of phenolic compounds and increased acidity.	Precise temperature and humidity control, suitable for standardized processes.	Risk of burnt flavors if temperature is too high.
Flatbed Dryer	Perforated surface where hot air is forced through a cocoa layer.	Drying time of 48–72 h, medium-high capacity.	Moderate energy consumption, depends on design.	Lower degradation of phenols at lower temperatures.	Simple, suitable for medium-scale production.	Requires frequent turning for uniform drying.
Rotary/Drum Dryer	Rotating drum where cocoa is mixed while receiving hot air.	Drying time of 12–24 h, high capacity.	High thermal efficiency, better drying uniformity.	Reduction of phenolic compounds at high temperatures.	Fast drying, prevents moisture accumulation.	High costs, risk of degradation at high temperatures.
Tunnel Dryer	Conveyor belts or movable trays in a tunnel with circulating hot air.	Drying time of 24–36 h, high capacity.	High energy consumption but high uniformity.	Good retention of polyphenols at controlled temperatures.	Allows control of drying variables, high efficiency.	Expensive, not suitable for small-scale producers.
Heat Pump Dryer	Closed-cycle dehumidified air system with heat recovery.	Drying time of 48–72 h, variable capacity.	Superior energy efficiency, heat reuse.	High retention of phenolic compounds and minimal impact on flavor.	Minimal impact on quality, better aroma preservation.	High installation cost, less accessible in rural areas.
Fluidized Bed Dryer	Hot air makes the beans float, allowing uniform drying.	Drying time of 12–24 h, medium capacity.	High thermal efficiency, rapid heat exchange.	May affect bean texture if parameters are not properly adjusted.	Fast and uniform, prevents localized overheating.	High cost and greater operational complexity.

**Table 3 foods-14-00721-t003:** Research on mathematical modeling in cocoa drying.

Research Objective	Key Assumptions	Experimental Conditions	Model Type(s)	Key Findings
Predicting influence of air velocity and water uniformity	Rate of contraction directly proportional to concentration; shrinkage factor considered constant; uniform initial water distribution; ellipsoidal bean geometry	Air temp: 30–60 °C; Air velocity: 0.3–1 m/s; Thickness: 0.5 mm	Isothermal, Moving Boundary, 3-D Ellipsoidal	Accurately predicts moisture content evolution, shrinkage, and changes in principal axes and testa thickness during drying [66].
Drying kinetics	Thin layer geometry	Air temp: 60, 75, 85 °C; Air velocity: 1.8, 2.7, 3.8 m/s	Newton’s	Drying rate increases with temperature [62]
Analyzing water and acetic acid migration	Local equilibrium among different forms; no chemical reaction; gaseous phase considered perfect gas	Air temp: 40 °C; Air velocity: 0.5 m/s; Relative moisture: 85%	Unidirectional Diffusion	Decrease in water and acetic acid diffusion coefficient with moisture content; crust formation observed at low moisture [83]
Evaluating polyphenol degradation	Spherical geometry; air temperature considered constant	Air temperature unspecified	1D unsteady-state	Agreement between model and experimental results [84]
Cocoa drying kinetics and water loss	Constant rate considered for momentum balance; energy changes due to air conduction effects; porosity and specific heat considered constant; initial moisture uniform	Air velocity: 3 m/s; Cocoa cultivar: CCN51; Dryer type: tunnel	Numerical solution using Finite Volume Method with exponential decay; Runge-Kutta method	Best match occurs at 40–50 °C; validation performed using experimental data; average absolute deviations reported [84]
Analyzing drying kinetics of cocoa beans using vacuum oven	Fuzzy logic model using 13 rules and 10 epochs for training	Air temperature: 40, 50, 60 °C; pressure: 245, 490, 735 mmHg	Fuzzy Logic	Model shows significant improvement in drying kinetics and quality (color, pH). Close match between theoretical and experimental results [85]
Investigating kinetics of drying and polyphenol degradation	First-order reaction assumed for polyphenol degradation; convective drying	Air temperature: 60, 70, 80 °C; Time: 24 h; Constant temperature	Convective	Empirical coefficients used; drying rate and polyphenol degradation determined [78]
Modeling cocoa drying with chemical reaction	Enzymatic oxidation and non-enzymatic condensation reactions of polyphenols considered	Air temp: 40, 50, 60 °C; Air velocity: 1 m/s; Relative moisture: 50, 60, 70, 80%	Luikov drying model combined with Fick’s diffusion model and first-order reaction rate equation	Polyphenol degradation and moisture diffusivity determined; model fit experimental data with low residual errors [86]
Modeling convective drying of fermented Amazonian cocoa beans	Bean assumed cylindrical shape; properties considered homogeneous; water transport radial; distribution axisymmetric	Air temp: 30, 40, 50, 60 °C; Air velocity: 0.3, 0.6, 1 m/s	Semi-theoretical model based on bean physics	Water diffusion equation, desorption isotherm, heat/mass transport equations used; model fit experimental data using least squares [65]
Predicting drying rate of cocoa beans in hot air dryer	Mixed-mode passive solar drier and direct solar drier;	Temperature: 60, 70, 80 °C; humidity: 2.9, 4.7, 6%	Modified Henderson and Pabis equation	Model successfully tested at all temperatures; two-tail test confirms suitability [64]
Modeling thin layer drying kinetics of cocoa beans using passive solar dryer	Mixed-mode passive solar dryer and open-sun drier; empirical coefficients added	Drying time: 14 h	Mixed-mode	Model satisfactorily explains drying behavior; moisture ratios compared and predicted values correlate well [87]
Characterizing thermal behavior of solar air collectors	Newtonian fluid in steady state and turbulent flow; viscous and turbulent dissipation considered; thermal radiation contribution ignored	Irradiation: 30–800 W/m^2^; Air mass flow: 0.009 to 0.026 kg/s	CFD	Model accurately predicts mass, momentum, and energy transport; three-step B-type collector is 67% more thermally efficient than one-step collector [88]

## Data Availability

No new data were created or analyzed in this study.

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
