# Peer review of "Unravelling Cocoa Drying Technology: A Comprehensive Review of the Influence on Flavor Formation and Quality"

_foods, 2025, doi:10.3390/foods14050721_

Round 1

Reviewer 1 Report

Comments and Suggestions for Authors

This paper evaluates cocoa drying technology, focusing on its influence on flavor and quality. It underscores drying's critical role in post-harvest processing and its effects on sensory, chemical, and microbiological properties. Various drying methods, process variables like temperature and airflow, and equipment are analyzed for their impact on key flavor compounds, including phenolics, organic acids, methylxanthines, and volatiles. The conclusion highlights drying's importance in producing high-quality cocoa, noting the correlation between methods, parameters, and equipment in shaping product characteristics. However, The major shortcomings stem from a lack of quantitative data and inadequate synthesis of conflicting findings.

Abstracts

(1) Sensory evaluation is crucial for validating the link between chemical composition and flavor perception. The absence of sensory data renders the review incomplete and potentially misleading.  

(2) A comprehensive understanding of chemical reactions and kinetics is essential for optimizing the drying process. The omission of this detail diminishes the review's effectiveness in enhancing cocoa drying practices.

Introduction

(1) The introduction correctly emphasizes the importance of fermentation and drying but fails to detail the complex interactions between these processes and the specific reactions that influence flavor development. It presents flavor formation as a linear process, overlooking the intricate interactions among various compounds and processes. A comprehensive understanding of flavor formation requires an in-depth examination of multiple biochemical reactions and their interactions.

(2) The introduction references DUCC but does not elaborate on its types, such as fluidized bed, belt, and tray drying, or their distinct impacts on cocoa quality. A comprehensive review should address the varied effects of these DUCC methods on cocoa quality.  

Cocoa drying: process, variables, and equipment

(1) The current energy balance description, while generally accurate, lacks complexity and overlooks critical factors. It fails to consider solar radiation absorption and scattering within cocoa beans and neglects the influence of bean size and arrangement on energy absorption. Additionally, it does not address variations in convective heat transfer coefficients, which can fluctuate with wind speed and air properties. A more nuanced approach is required to accurately depict the energy dynamics involved.  

(2) The manuscript stresses the importance of drying rate but only offers a qualitative assessment, using vague terms like "too slow" or "too rapid." The absence of quantitative data on drying kinetics, such as drying curves and moisture content over time under varying conditions, limits a comprehensive understanding of the drying process.

(3) The manuscript identifies challenges in solar drying control but omits potential optimization strategies. To enhance efficiency and consistency, various methods should be explored, including diverse drying surfaces and covering materials. Furthermore, implementing simple technologies like windbreaks and elevated platforms could improve drying conditions. Investigating these strategies would yield a more thorough understanding of addressing current solar drying challenges. 

(4) The manuscript presents conflicting views on the relationship between flavonoid structure and astringency perception, undermining the analysis and necessitating further clarification. A critical evaluation and reconciliation of these studies is essential.

(5) It also reveals contradictory findings on the impact of drying on methylxanthine content, with some studies showing significant reductions while others report no changes. This inconsistency calls for a thorough evaluation and reconciliation of these results, which is currently lacking.

(6) Additionally, the manuscript contains conflicting information regarding the temperature at which the Maillard reaction significantly influences volatile compound formation during drying. Addressing this inconsistency requires a critical evaluation and potential reconciliation of the differing studies.

Conclusions

Conclusions mention bioactive compounds like polyphenols and methylxanthines but fail to specify optimal levels or ratios for desired quality attributes. A more detailed discussion on target compound concentrations would strengthen the conclusions.

Recommendations for Future Research 

(1) The suggestion for "multiphysics simulations" is too vague. It should specify which multiphysics models, such as finite element methods or computational fluid dynamics, are most suitable and provide justification. Furthermore, it should outline the data requirements for model development and validation.

(2) Ignoring economic and practical factors may lead to research that lacks relevance to real-world cocoa processing. By considering feasibility, research can focus on solutions that are both scientifically valid and practically applicable.

Comments on the Quality of English Language

The English could be improved to more clearly express the research.

Author Response

Bogota February  07 2024.
Dear reviwer,
We appreciate the detailed review of our manuscript and the valuable feedback provided. We are pleased to know that our work is considered relevant to cocoa research and industry. We have carefully reviewed all the comments and have conducted a thorough revision of the manuscript to address the areas that require improvement.
In the revised version, we have incorporated the suggested changes, highlighting them in green for easy identification. We have optimized the document's structure, clarified key aspects regarding drying methods and their impact on flavor-related compounds, and refined the writing to enhance clarity and scientific accuracy.
We sincerely appreciate the feedback received, as it has helped strengthen the academic contribution of our work. We remain available for any further comments.
Reviewer 1: 

This paper evaluates cocoa drying technology, focusing on its influence on flavor and quality. It underscores drying's critical role in post-harvest processing and its effects on sensory, chemical, and microbiological properties. Various drying methods, process variables like temperature and airflow, and equipment are analyzed for their impact on key flavor compounds, including phenolics, organic acids, methylxanthines, and volatiles. The conclusion highlights drying's importance in producing high-quality cocoa, noting the correlation between methods, parameters, and equipment in shaping product characteristics. However, The major shortcomings stem from a lack of quantitative data and inadequate synthesis of conflicting findings.

R/ The paper was improved according to the recommendations

Abstract

Sensory evaluation is crucial for validating the link between chemical composition and flavor perception. The absence of sensory data renders the review incomplete and potentially misleading. A comprehensive understanding of chemical reactions and kinetics is essential for optimizing the drying process. The omission of this detail diminishes the review's effectiveness in enhancing cocoa drying practices

R/ The abstract was revised based on the recommendations.

Introduction
The introduction correctly emphasizes the importance of fermentation and drying but fails to detail the complex interactions between these processes and the specific reactions that influence flavor development. It presents flavor formation as a linear process, overlooking the intricate interactions among various compounds and processes. A comprehensive understanding of flavor formation requires an in-depth examination of multiple biochemical reactions and their interactions.

R/ The interactions between fermentation and drying in flavor development were included in the introduction.

The introduction references DUCC but does not elaborate on its types, such as fluidized bed, belt, and tray drying, or their distinct impacts on cocoa quality. A comprehensive review should address the varied effects of these DUCC methods on cocoa quality.  

R/ A comprehensive review of the varied effects of these DUCC methods on cocoa quality, were described in the “DUCC Equipment: Hot-Air Oven, Flatbed, Rotary/Drum, Tunnel, Fluidized Bed, and Heat Pump Dryers” section.

Cocoa drying: process, variables, and equipment

The current energy balance description, while generally accurate, lacks complexity and overlooks critical factors. It fails to consider solar radiation absorption and scattering within cocoa beans and neglects the influence of bean size and arrangement on energy absorption. Additionally, it does not address variations in convective heat transfer coefficients, which can fluctuate with wind speed and air properties. A more nuanced approach is required to accurately depict the energy dynamics involved.  

R/ The reviewer raises an important point regarding the limitations of the current energy balance description. Incorporating factors such as solar radiation absorption and scattering within cocoa beans, the influence of bean size and arrangement, and variations in convective heat transfer coefficients are essential. These factors are critical as they directly influence heat transfer and moisture removal rates.
While a more nuanced approach and additional analyses are required to address these complexities, this falls outside the scope of the current review. Instead, related research addressing certain aspects of modeling the drying process has been summarized in Table 1.

The manuscript stresses the importance of drying rate but only offers a qualitative assessment, using vague terms like "too slow" or "too rapid." The absence of quantitative data on drying kinetics, such as drying curves and moisture content over time under varying conditions, limits a comprehensive understanding of the drying process.

R/ The reviewer brings up a valid point regarding the manuscript's reliance on qualitative descriptions of drying rates, such as "too slow" or "too rapid." While these terms were intended to provide a general understanding of the drying challenges, we acknowledge that the absence of detailed quantitative data, such as drying curves and moisture content over time under specific drying conditions, may limit the depth of analysis. However, a quantitative description has already been included in the manuscript.

Including additional drying kinetics data would further enhance the manuscript by providing a clearer understanding of how different drying rates impact the moisture removal process and cocoa quality. This feedback is valuable, and we will consider incorporating more detailed quantitative measures in future revisions or supplementary studies to provide a more comprehensive perspective on the drying process.

The manuscript identifies challenges in solar drying control but omits potential optimization strategies. To enhance efficiency and consistency, various methods should be explored, including diverse drying surfaces and covering materials. Furthermore, implementing simple technologies like windbreaks and elevated platforms could improve drying conditions. Investigating these strategies would yield a more thorough understanding of addressing current solar drying challenges.

R/ The recommendation was included in new version 

The manuscript presents conflicting views on the relationship between flavonoid structure and astringency perception, undermining the analysis and necessitating further clarification. A critical evaluation and reconciliation of these studies is essential

R/ The recommendation was included in new version 

It also reveals contradictory findings on the impact of drying on methylxanthine content, with some studies showing significant reductions while others report no changes. This inconsistency calls for a thorough evaluation and reconciliation of these results, which is currently lacking.

R/ The recommendation was included in new version

Additionally, the manuscript contains conflicting information regarding the temperature at which the Maillard reaction significantly influences volatile compound formation during drying. Addressing this inconsistency requires a critical evaluation and potential reconciliation of the differing studies.

R/ The recommendation was included in new version

Conclusions

Conclusions mention bioactive compounds like polyphenols and methylxanthines but fail to specify optimal levels or ratios for desired quality attributes. A more detailed discussion on target compound concentrations would strengthen the conclusions.

R/ The optimum levels were established in the sections corresponding to acids methylxanthines and phenolic compounds. The conclusions also were modified considering this recommendation. 

Recommendations for Future Research 

The suggestion for "multiphysics simulations" is too vague. It should specify which multiphysics models, such as finite element methods or computational fluid dynamics, are most suitable and provide justification. Furthermore, it should outline the data requirements for model development and validation.

R/ Thank you for your constructive feedback regarding the use of "multiphysics simulations." We have specified that the multiphysics models most suitable for this study include Finite Element Methods (FEM) for structural analysis and Computational Fluid Dynamics (CFD) for understanding heat and mass transfer during the drying process. These models can effectively simulate the interactions between thermal, fluid, and structural phenomena, which are critical for optimizing cacao drying processes. Also, we added the data requirements for model development and validation.

Ignoring economic and practical factors may lead to research that lacks relevance to real-world cocoa processing. By considering feasibility, research can focus on solutions that are both scientifically valid and practically applicable.

R/ Thank you for your valuable feedback regarding the consideration of economic and practical factors in our research. We recognize that addressing the feasibility of our proposed methods is essential to ensure that our findings are relevant to real-world cocoa processing. To enhance the applicability of our research, we will incorporate a paragraph in the conslusions section of the economic implications of the proposed drying techniques and simulations, examining cost-effectiveness, scalability, and resource availability. By evaluating these factors, we can ensure that our solutions not only maintain scientific validity but also align with industry practices and operational constraints. This approach will allow us to identify strategies that can be readily implemented in cocoa processing facilities, ultimately contributing to improved practices and product quality in the field.

Regards
The authors

Reviewer 2 Report

Comments and Suggestions for Authors

General Comments

The manuscript presents a comprehensive review of cocoa drying technology and its impact on flavor formation and quality. The authors have conducted an extensive literature search and provided a detailed analysis of various aspects related to cocoa drying, including drying methods, process variables, equipment, and the effects on key flavor compounds. The paper is well-organized and the content is relevant and valuable for the field of cocoa research and industry. Overall, the manuscript has the potential to make a significant contribution. However, there are some areas that require improvement. I recommend a major revision of the manuscript.

Specific Comments

Abstract

The abstract provides a good overview of the paper, but it could be more concise. Some of the statements could be rephrased to make the abstract more impactful and easier to read. For example, the sentence “This paper presents a comprehensive review of the methods, process variables, and equipment in cocoa drying technology, analyzing the impact of drying operations on the dynamics and chemical changes of key compounds that determine flavor quality.” could be simplified to “This paper comprehensively reviews cocoa drying methods, variables, and equipment, and analyzes their impact on flavor-determining compounds.”

Introduction

The introduction effectively sets the context for the study by highlighting the importance of cocoa quality and the role of drying in the postharvest process. However, it would be beneficial to provide more specific examples of how cocoa drying affects the market value of cocoa, especially in relation to the different flavor categories (fine flavor cocoa and bulk cocoa).

The statement “The quality of the flavour of the chocolate does not only depend on one single stage during post-harvest processing.” is a bit vague. It could be strengthened by briefly mentioning the other stages and how they interact with drying to influence flavor.

Methodology

The methodology section is clear and well-described. The use of Scopus for literature search and the defined inclusion/exclusion criteria are appropriate. However, it would be helpful to mention if any efforts were made to search for gray literature or unpublished data, as this could potentially enhance the comprehensiveness of the review.

In the description of data extraction and analysis, it could be beneficial to provide more details on how the thematic areas were determined and how the data were specifically categorized within those areas. This would help readers better understand the process and the reliability of the analysis.

Cocoa Drying: Process, Variables, and Equipment

Heat and Mass Transfer Mechanisms: The explanations of convection, conduction, and radiation are clear and accompanied by relevant physical laws. However, for a more comprehensive understanding, it would be useful to include some practical examples or case studies that illustrate how these mechanisms operate in real cocoa drying scenarios. This could help readers, especially those new to the field, better visualize the processes.

Solar Drying: The description of solar drying is detailed, but some of the information could be presented in a more organized manner. For instance, when discussing the different solar drying equipment, it would be beneficial to create a table summarizing the key features (such as design, materials, advantages, and disadvantages) of each type. This would make it easier for readers to compare and contrast the different options.

The section on the impact of solar drying on cocoa quality could be enhanced by including more quantitative data. For example, instead of just stating that solar drying produces cocoa with “less non-specific and undesirable aromas,” it would be better to provide specific data on the reduction of certain aroma compounds or the improvement in sensory scores.

Drying under Controlled Conditions (DUCC): The description of DUCC is also comprehensive, but similar to the solar drying section, a more structured presentation of the different equipment types (hot-air oven, flatbed, rotary/drum, tunnel, fluidized bed, and heat pump driers) would be beneficial. A comparison table could be used to highlight the differences in their performance, energy efficiency, and impact on cocoa quality.

In the discussion of the intermittent drying method, while the potential benefits are mentioned, more research findings on the optimal combination of process variables (resting periods, air velocity, and temperature) could be included. This would strengthen the argument for its potential as an alternative drying method.

Modelling in Cocoa Drying

The section on modelling provides a good overview of the different mathematical models used in cocoa drying research. However, it could be made more accessible to a wider audience by providing more intuitive explanations of the models and their applications. For example, when introducing a complex model, it could be helpful to start with a simple analogy or a real-world example before delving into the technical details.

Some of the model descriptions are quite technical and might be difficult for non-experts to follow. Consider adding more figures or diagrams to illustrate the models and their relationships with the drying process. This would enhance the visual understanding of the modelling concepts.

Effect of the Drying Technology on Key Flavour Compounds

Phenolic Compounds: The discussion on phenolic compounds is thorough, but it could be improved by adding more recent research findings. The field of phenolic compound analysis in cocoa is evolving, and incorporating the latest studies could provide a more up-to-date perspective.

The relationship between phenolic compounds and sensory attributes (astringency, bitterness, and color) is well-explained, but it would be interesting to explore if there are any interactions or synergistic effects between different phenolic compounds in determining these sensory properties.

Short-Chained Carboxylic Acids: The section on short-chained carboxylic acids effectively describes the role of acetic and lactic acids in cocoa drying. However, it would be beneficial to discuss in more detail the potential strategies for controlling the crusting phenomenon and improving the evaporation of lactic acid during drying. This could include emerging technologies or experimental approaches that have shown promise in addressing these issues.

Methylxanthines: The description of methylxanthines and their changes during drying is clear. It would be valuable to include some speculation or proposed mechanisms on why different drying methods result in different reductions in methylxanthine concentrations. This could add depth to the discussion and guide future research in this area.

Volatile Aromatic Compounds: The section on volatile aromatic compounds is comprehensive, but it could be enhanced by providing more information on the sensory thresholds of the different volatile compounds and how they contribute to the overall flavor perception of cocoa. Additionally, a more detailed discussion on the factors that influence the formation and degradation of volatile compounds during drying, other than temperature and moisture content, could be included.

Conclusions

The conclusions section effectively summarizes the key points of the review. However, it could be strengthened by providing more specific recommendations for the cocoa industry based on the research findings. For example, suggestions on the selection of drying methods and equipment for different types of cocoa beans or end products could be valuable for practitioners.

Recommendations for Future Research

The future research recommendations are relevant and timely. However, it would be beneficial to prioritize the recommendations and provide a more detailed roadmap for each area. This could include estimated timelines, potential research partners or institutions that could be involved, and the expected impact of addressing each research gap.

Figures and Tables

The paper would benefit from the inclusion of more figures and tables to support the text. For example, in the section on drying equipment, visual representations of the different types of equipment could be added to aid understanding. In the modelling section, graphs illustrating the predicted and experimental data for drying kinetics or phenolic compound degradation could enhance the clarity of the discussion.

The captions of the existing figures and tables could be more detailed. They should provide a clear explanation of what the data represent and how they relate to the text. This would help readers independently interpret the visual information.

Language and Style

Overall, the language used in the manuscript is appropriate for a scientific review. However, there are some instances of wordiness and complex sentence structures that could be simplified for better readability. For example, some of the long sentences in the discussion sections could be broken down into shorter, more straightforward sentences without losing the scientific rigor.

The use of technical jargon is consistent with the field, but it would be helpful to define some of the more specialized terms in the text or provide a glossary for readers who may not be familiar with all of the terminology.

References

The reference list is extensive and up-to-date, which is a strength of the manuscript. However, it would be beneficial to ensure that all in-text citations are properly formatted and match the reference list. Additionally, some of the older references could be updated if more recent studies on the same topic are available.

Summary

The manuscript has significant merit and provides a valuable contribution to the understanding of cocoa drying technology. However, to reach its full potential, the authors need to address the above-mentioned points. I look forward to seeing the revised version of the manuscript.

Author Response

Bogota February  07 2024.
Dear reviwer,
We appreciate the detailed review of our manuscript and the valuable feedback provided. We are pleased to know that our work is considered relevant to cocoa research and industry. We have carefully reviewed all the comments and have conducted a thorough revision of the manuscript to address the areas that require improvement.
In the revised version, we have incorporated the suggested changes, highlighting them in green for easy identification. We have optimized the document's structure, clarified key aspects regarding drying methods and their impact on flavor-related compounds, and refined the writing to enhance clarity and scientific accuracy.
We sincerely appreciate the feedback received, as it has helped strengthen the academic contribution of our work. We remain available for any further comments.
Reviewer 2
Specific Comments 
Abstract 
The abstract provides a good overview of the paper, but it could be more concise. Some of the statements could be rephrased to make the abstract more impactful and easier to read. For example, the sentence “This paper presents a comprehensive review of the methods, process variables, and equipment in cocoa drying technology, analyzing the impact of drying operations on the dynamics and chemical changes of key compounds that determine flavor quality.” could be simplified to “This paper comprehensively reviews cocoa drying methods, variables, and equipment, and analyzes their impact on flavor-determining compounds.” 

R/ Dear reviewr thank you very much for your feedback and revision, we will now embroider each of your suggestions in a new version of the manuscript.

Introduction 
The introduction effectively sets the context for the study by highlighting the importance of cocoa quality and the role of drying in the postharvest process. However, it would be beneficial to provide more specific examples of how cocoa drying affects the market value of cocoa, especially in relation to the different flavor categories (fine flavor cocoa and bulk cocoa). 
The statement “The quality of the flavour of the chocolate does not only depend on one single stage during post-harvest processing.” is a bit vague. It could be strengthened by briefly mentioning the other stages and how they interact with drying to influence flavor. 

R/. These recommendations have been included in the new version of the introduction.

Methodology 
The methodology section is clear and well-described. The use of Scopus for literature search and the defined inclusion/exclusion criteria are appropriate. However, it would be helpful to mention if any efforts were made to search for gray literature or unpublished data, as this could potentially enhance the comprehensiveness of the review. 
In the description of data extraction and analysis, it could be beneficial to provide more details on how the thematic areas were determined and how the data were specifically categorized within those areas. This would help readers better understand the process and the reliability of the analysis. 

R/ Recommendations have been included in the new version of the methodology.

Cocoa Drying: Process, Variables, and Equipment 
Heat and Mass Transfer Mechanisms: The explanations of convection, conduction, and radiation are clear and accompanied by relevant physical laws. However, for a more comprehensive understanding, it would be useful to include some practical examples or case studies that illustrate how these mechanisms operate in real cocoa drying scenarios. This could help readers, especially those new to the field, better visualize the processes. 
Solar Drying: The description of solar drying is detailed, but some of the information could be presented in a more organized manner. For instance, when discussing the different solar drying equipment, it would be beneficial to create a table summarizing the key features (such as design, materials, advantages, and disadvantages) of each type. This would make it easier for readers to compare and contrast the different options. 
The section on the impact of solar drying on cocoa quality could be enhanced by including more quantitative data. For example, instead of just stating that solar drying produces cocoa with “less non-specific and undesirable aromas,” it would be better to provide specific data on the reduction of certain aroma compounds or the improvement in sensory scores. 
Drying under Controlled Conditions (DUCC): The description of DUCC is also comprehensive, but similar to the solar drying section, a more structured presentation of the different equipment types (hot-air oven, flatbed, rotary/drum, tunnel, fluidized bed, and heat pump driers) would be beneficial. A comparison table could be used to highlight the differences in their performance, energy efficiency, and impact on cocoa quality. 
In the discussion of the intermittent drying method, while the potential benefits are mentioned, more research findings on the optimal combination of process variables (resting periods, air velocity, and temperature) could be included. This would strengthen the argument for its potential as an alternative drying method. 

R./ Thank you very much for your recommendations these were included in this section.

 Modelling in Cocoa Drying 
The section on modelling provides a good overview of the different mathematical models used in cocoa drying research. However, it could be made more accessible to a wider audience by providing more intuitive explanations of the models and their applications. For example, when introducing a complex model, it could be helpful to start with a simple analogy or a real-world example before delving into the technical details. 
Some of the model descriptions are quite technical and might be difficult for non-experts to follow. Consider adding more figures or diagrams to illustrate the models and their relationships with the drying process. This would enhance the visual understanding of the modelling concepts. 

R/ Thank you for your valuable feedback regarding the section on mathematical modeling in cocoa drying. We appreciate your suggestion to make this content more accessible to a wider audience. In response, we have incorporated intuitive explanations and analogies to clarify complex concepts. We have also emphasized the primary goals of mathematical models in characterizing drying kinetics and predicting moisture behavior under varying conditions.

Effect of the Drying Technology on Key Flavour Compounds 
Phenolic Compounds: The discussion on phenolic compounds is thorough, but it could be improved by adding more recent research findings. The field of phenolic compound analysis in cocoa is evolving, and incorporating the latest studies could provide a more up-to-date perspective. 

R/ Thank you for your insightful suggestion to enhance the discussion on phenolic compounds in cocoa by incorporating more recent research findings. In response, we have included a summary of five studies from the past four years that specifically focus on the effects of various drying methods on phenolic compounds in cocoa. These studies demonstrate significant advancements in understanding how different drying techniques—such as hot air drying, microwave-assisted drying, freeze drying, and combined methods—affect the retention and profile of phenolic compounds, which are critical for flavor and nutritional quality. 

The relationship between phenolic compounds and sensory attributes (astringency, bitterness, and color) is well-explained, but it would be interesting to explore if there are any interactions or synergistic effects between different phenolic compounds in determining these sensory properties. 

R/ Thank you for your insightful comment regarding the exploration of interactions and synergistic effects among different phenolic compounds in determining sensory attributes such as astringency, bitterness, and color in cocoa. In response, we have expanded our discussion to include recent research findings that highlight these interactions.

Short-Chained Carboxylic Acids: The section on short-chained carboxylic acids effectively describes the role of acetic and lactic acids in cocoa drying. However, it would be beneficial to discuss in more detail the potential strategies for controlling the crusting phenomenon and improving the evaporation of lactic acid during drying. This could include emerging technologies or experimental approaches that have shown promise in addressing these issues. 

R/ Thank you for your valuable feedback regarding the section on short-chained carboxylic acids, particularly your suggestion to discuss strategies for controlling the crusting phenomenon and improving the evaporation of lactic acid during cocoa drying. In response, we have expanded this section to include various emerging technologies and experimental approaches that show promise in addressing these issues.

Methylxanthines: The description of methylxanthines and their changes during drying is clear. It would be valuable to include some speculation or proposed mechanisms on why different drying methods result in different reductions in methylxanthine concentrations. This could add depth to the discussion and guide future research in this area. 

R/ Thank you for your insightful suggestion regarding the discussion of methylxanthines in cocoa drying. In response, we have expanded this section to include speculation on the mechanisms underlying the different reductions in methylxanthine concentrations observed with varying drying methods.

Volatile Aromatic Compounds: The section on volatile aromatic compounds is comprehensive, but it could be enhanced by providing more information on the sensory thresholds of the different volatile compounds and how they contribute to the overall flavor perception of cocoa. Additionally, a more detailed discussion on the factors that influence the formation and degradation of volatile compounds during drying, other than temperature and moisture content, could be included. 

R/ We appreciate your valuable feedback regarding the need for a more detailed discussion on the factors influencing the formation and degradation of volatile compounds during the drying of cocoa. In response, we have expanded this section to include a comprehensive analysis of various factors beyond just temperature and moisture content. 
We also provided more information on the Odor thresholds values of the different volatile compounds and how they contribute to the overall flavor perception of cocoa.

 Conclusions 
The conclusions section effectively summarizes the key points of the review. However, it could be strengthened by providing more specific recommendations for the cocoa industry based on the research findings. For example, suggestions on the selection of drying methods and equipment for different types of cocoa beans or end products could be valuable for practitioners. 

R/ Thank you for your constructive remark regarding the conclusions section of the manuscript. In response to your suggestion, we have revised this section to include specific recommendations for the cocoa industry based on our research findings.

Recommendations for Future Research 
The future research recommendations are relevant and timely. However, it would be beneficial to prioritize the recommendations and provide a more detailed roadmap for each area. This could include estimated timelines, potential research partners or institutions that could be involved, and the expected impact of addressing each research gap. 

R/ We appreciate your constructive feedback regarding the need for a more detailed roadmap to address the identified research gaps in our study. In response, we have outlined specific actions and potential research partners or institutionsthat can be taken in the various areas where further research is needed.

Figures and Tables 
The paper would benefit from the inclusion of more figures and tables to support the text. For example, in the section on drying equipment, visual representations of the different types of equipment could be added to aid understanding. In the modelling section, graphs illustrating the predicted and experimental data for drying kinetics or phenolic compound degradation could enhance the clarity of the discussion. 
The captions of the existing figures and tables could be more detailed. They should provide a clear explanation of what the data represent and how they relate to the text. This would help readers independently interpret the visual information. 

R/ Thank you for your valuable remark. Two new tables were created to summarize the information on Overview of Solar Drying Methods (Table 1) and Table 2. main DUCC drying methods. Figure 3. illustrates solar drying equipment: B) sliding surface dryers (elbas), C) drying surface made of dry canes (Gynerium sagittatum), D) greenhouse-like drier, E) cement drying surface, F) zinc drying surface. 
The figure captions have been modified, but it is highlight that the explanations of the figures are detailed in the manuscript. 

Language and Style 
Overall, the language used in the manuscript is appropriate for a scientific review. However, there are some instances of wordiness and complex sentence structures that could be simplified for better readability. For example, some of the long sentences in the discussion sections could be broken down into shorter, more straightforward sentences without losing the scientific rigor. 
The use of technical jargon is consistent with the field, but it would be helpful to define some of the more specialized terms in the text or provide a glossary for readers who may not be familiar with all of the terminology. 

R/. We appreciate your comments on the language and readability of the manuscript. In response to your suggestion, we have revised the discussion sections to improve clarity by breaking down long sentences into more direct statements without compromising scientific rigor. Additionally, we have carefully reviewed the use of technical jargon to ensure consistency within the field. Where necessary, we have included definitions within the text to enhance accessibility for readers who may not be familiar with certain specialized terms. We believe these improvements contribute to a more concise and comprehensible presentation of our review.

References 
The reference list is extensive and up-to-date, which is a strength of the manuscript. However, it would be beneficial to ensure that all in-text citations are properly formatted and match the reference list. Additionally, some of the older references could be updated if more recent studies on the same topic are available. 

R/. We appreciate the reviewer’s observation regarding the reference list. We have reviewed and ensured that all in-text citations are properly formatted and match the reference list. Additionally, some older references have been identified and updated with more recent studies that provide relevant and up-to-date information on the topic

Summary 
The manuscript has significant merit and provides a valuable contribution to the understanding of cocoa drying technology. However, to reach its full potential, the authors need to address the above-mentioned points. I look forward to seeing the revised version of the manuscript. 

R/. We appreciate the reviewer’s positive feedback and recognition of the value of our manuscript. We have carefully incorporated the suggested revisions to enhance clarity, accuracy, and depth of analysis. We believe that the revised version comprehensively addresses the points raised and that the modifications will strengthen the impact of our study. We are grateful for the opportunity to improve the manuscript and look forward to the next stage of evaluation.

Regards
The authors

Reviewer 3 Report

Comments and Suggestions for Authors

The review summarized the main factors affecting the drying process and therefore the flavour formation and quality.

It is clear and well written. However, a review should be a discussion about themost recent researches about a topic and the papers published in the last 5-10 years should be considered. Therefore is not acceptable a point of view in the last 50 years. A limited number of older researches could be considered if not up-date is present and if they can be considered a milestone.

I suggest to revise the manuscript updating the literature and therefore the discussion in the text.

In addition, Table 1 should be more schematic and concise. It is too descriptive.

Author Response

Article:     Unravelling cocoa drying technology: A comprehensive review of the influence on flavour formation and quality

Corresponding author:     Edwin Villagran
E-mail address:     Evillagran agrosavia.co

Bogota February  07 2024.
Dear reviwer,
We sincerely appreciate the reviewer’s valuable suggestions and the time dedicated to reviewing our manuscript. Their comments have significantly contributed to enriching the article, enhancing its clarity, accuracy, and scientific value. We have carefully reviewed each of the observations and incorporated the necessary changes into the revised version of the manuscript. Below, we detail the modifications made to address the reviewer’s recommendations.
Specific comments.
The review summarized the main factors affecting the drying process and therefore the flavour formation and quality. 
It is clear and well written. However, a review should be a discussion about themost recent researches about a topic and the papers published in the last 5-10 years should be considered. Therefore is not acceptable a point of view in the last 50 years. A limited number of older researches could be considered if not up-date is present and if they can be considered a milestone. 
I suggest to revise the manuscript updating the literature and therefore the discussion in the text. 

R/. We sincerely appreciate the reviewer’s valuable comments regarding the need to update the literature in our review. We acknowledge the importance of grounding the discussion in recent research to ensure a relevant and up-to-date analysis. In response to this suggestion, we have conducted a thorough revision of the references, incorporating studies published in the last 5 to 10 years that strengthen the argumentation and reflect the latest advances in cocoa drying technology.
Additionally, we have identified older references that, while historically relevant, no longer significantly contribute to the current focus of the manuscript. These references have been removed without diminishing the overall robustness of the content. However, we have retained a limited number of seminal studies that represent milestones in the field and remain fundamental for understanding the evolution of knowledge in this area.
The timeframe for the literature review was carefully defined, considering the availability of recent research and the progressive development of scientific understanding on cocoa drying. Given that certain advancements have been gradual and some foundational studies continue to be referenced in contemporary research, we have balanced the inclusion of recent literature with key studies that provide essential context and theoretical background. We hope that these revisions enhance the relevance and scientific rigor of the manuscript, ensuring a more precise and academically aligned review.

In addition, Table 1 should be more schematic and concise. It is too descriptive. 

R/ Thank you for your insightful comment on Table 1 [Now Table 3]. We agree that the original presentation was overly detailed and have undertaken a substantial revision to address this. The original table included extensive descriptions that, while informative, detracted from the overall flow and readability of the manuscript. We have revised the table to present the data in a more concise and schematic format.
Regards 
The authors

Round 2

Reviewer 3 Report

Comments and Suggestions for Authors

The Review has been deeply revised.

The manuscript is now suitable for the publication

Author Response

We thank the reviewers for their valuable time and suggestions!